# Glycan-based biological degraders targeting the cytokine immune axis
Michelle Seifert[1], Tim Kollenkirchen[1], Andreas Ernst[1], Samaneh Rasoulinejad[1], Sarah M. S. Koellner[1], Ann-Kathrin Schneider[1], Marcin Luzarowski[2], Nicole Lübbehusen[2], Di Wu[1,3,4], Aimo Kannt [1,5,6] & Schara Safarian [1,3,4,5,6] ✉

Targeted degradation of extracellular proteins via endo-lysosomal pathways constitutes a promising therapeutic strategy by enabling irreversible target removal and potentially reducing systemic effects of antigen accumulation. In this study, we present a systematic approach for the design and evaluation of Biological Degraders (BioDegs) that leverage receptor-mediated uptake through triantennary N-acetylgalactosamine (TGN) induced activation of the asialoglycoprotein receptor (ASGPR). While this principle has been established, there is still a lack about molecular determinants that influence uptake efficiency. Using interleukin-6 (IL-6) and its soluble receptor (sIL-6R) as therapeutically relevant model antigens, we systematically compared a range of BioDeg formats, including full-length antibodies (siltuximab, tocilizumab), a VHH against IL-6, and an IL-6 binding decoy-receptor design, with respect to binding affinity, thermal stability, cellular uptake, lysosomal trafficking, and degradation efficiency in HepG2 cells. Our results demonstrate that combined properties of scaffold architecture and receptor engagement critically influence degradation outcomes. This work provides a comparative framework for BioDeg design and highlights key parameters for developing lysosome-targeting degraders for extracellular proteins.

A substantial part of the human proteome is associated with diseases such as cancer, neurodegenerative disorders, and autoimmune conditions[1,2]. Despite significant progress in drug development over recent decades, a large fraction of proteins remains "undruggable" due to challenges in accessibility or specificity[3,4]. One promising new therapeutic principle is proximity-induced targeted protein degradation (TPD). In contrast to conventional small-molecule drugs or biologics, molecular degraders (MDs) promote the clearance of disease-related proteins by exploiting innate protein homeostasis and quality control systems[5]. A fundamental distinction between molecular degraders and conventional inhibitors or neutralizing proteins is that MDs do not require to interact with functionally important sites to disrupt biochemical processes. Instead, degraders can operate by recognizing accessible surfaces of pathogenic proteins, regardless of their biochemical activity[5,6]

Current TPD technologies include proteolysis-targeting chimeras (PROTACs) and molecular glues, which use the ubiquitin-proteasome system to degrade intracellular proteins[7–9]. Alternative approaches such as AUTACs and ATTECs utilize the autophagy-lysosome pathway to degrade larger protein complexes or aggregates[10–12]. While these technologies center around the frameworks of small-molecules and act on intracellular target proteins, exploiting the endo-lysosomal pathway for TPD offers the possibility to address extracellular proteins and integral membrane proteins with extracellular domains via biologics such as antibodies. This concept was pioneered by Igawa et al., who demonstrated how pH-dependent binding of an engineered anti-IL-6R IgG1 can facilitate IL-6R degradation[13]. This sweeping antibody design principle results in antibody induced internalization and lysosomal degradation of the target antigen while the antibody is cycled back to the plasma membrane by maintaining interaction with Fc receptors.

An alternative biological degrader technology based on glycobiology principles was introduced by the Bertozzi laboratory. Lysosomal-targeting chimeras (LYTACs) are constructed from scaffolds of monoclonal antibodies decorated with either mannose-6-phosphonate (M6Pn) or triantennary N-acetylgalactosamine (TGN) moieties[14,15]. These glycans function

[1]Fraunhofer Institute for Translational Medicine and Pharmacology ITMP, Frankfurt, Germany. [2]Core Facility for Mass Spectrometry and Proteomics, Center for Molecular Biology of Heidelberg University (ZMBH), DKFZ-ZMBH Alliance, Heidelberg, Germany. [3]China-Singapore Joint Laboratory on Liver Disease Research, The First Hospital of Jilin University, Changchun, Jilin, China. [4]Department and Emeritus Group of Molecular Membrane Biology, Max Planck Institute of Biophysics, Frankfurt, Germany. [5]Fraunhofer Cluster of Excellence for Immune Mediated Diseases (CIMD), Frankfurt, Germany. [6]Institute of Clinical Pharmacology, Faculty of Medicine, Goethe University Frankfurt, Frankfurt, Germany. ✉e-mail: schara.safarian@itmp.fraunhofer.de

as specific ligands for lectin-type lysosomal-targeting receptors (LTRs), initiating target endocytosis and lysosomal trafficking through clathrin-dependent receptor-mediated endocytosis via the ubiquitous insulin-like growth factor receptor 2 (IGF2R) in case of M6Pn-LYTACs[16–20], or the hepatocyte-specific asialoglycoprotein receptor (ASGPR) for TGN-based LYTACs[21,22]. This novel degradation mechanism thus enables tissue-specific degradation of secreted and membrane-integrated proteins, or ubiquitous clearance of these respective protein classes, depending on the distribution of the selected LTR.

In recent years, the field of extracellular degraders has rapidly evolved and produced versatile design principles including aptamer, cyclic peptide, antibody, and small-molecule LTR ligands[14,23–26]. The use of generative AI for protein design has further enabled the development of miniproteins that can stabilize active-conformations of IGF2R or facilitate oligomerization of ASGPR in order to induce endocytosis and lysosomal trafficking[27]. Moreover, Baker and coworkers were able to design synthetic proteins that target alternative LTRs such as the multipurpose receptor sortilin (abundant in brain and liver tissue), and transferrin receptor (ubiquitous, highly abundant in proliferative tissue and the blood-brain barrier). Effective degradation of several clinically relevant membrane and extracellular proteins, including PD-L1, EGFR, TGF-β, and PDGF, has been demonstrated via the abovementioned extracellular degrader modalities[14,20,23,24].

Building on these advances, extracellular degraders represent a compelling strategy for targeting cytokines that act as central drivers of systemic and chronic disease. Interleukin-6 (IL-6) is a pleiotropic pro-inflammatory cytokine that plays critical roles in immune regulation, inflammation, hematopoiesis, and oncogenesis[28]. Dysregulated IL-6 signaling, either via classical cis-signaling or soluble IL-6 receptor (sIL-6R) facilitated trans-signaling, is implicated in acute and chronic inflammatory conditions such as rheumatoid arthritis (RA) and systemic lupus erythematosus (SLE)[29–31]. Moreover, IL-6 promotes tumor cell survival, angiogenesis, and immune evasion, primarily via activation of the STAT3 signaling pathway[32]. Consequently, therapeutic inhibition of IL-6 signaling has shown clinical benefit[33,34]. Currently approved biologics include tocilizumab and sarilumab (anti-IL-6R antibodies), as well as siltuximab (an IL-6 neutralizing antibody), used in the therapy of, amongst others, RA, Castleman disease, and COVID-19-related hyperinflammation[35–41]. Building on this clinical success, there is growing interest in extending targeted cytokine modulation beyond neutralization toward degradation strategies, particularly in immune-mediated diseases such as allergy, autoimmunity, and hematological malignancies. Such diseases often rely on cytokine signaling and current biologics targeting IL-6, TNF-α (e.g., infliximab, adalimumab), IL-1β (canakinumab), or IL-17 (secukinumab) have shown therapeutic benefit, yet response rates remain variable[42–48]. For example, IL-6 inhibition in Castleman disease improves inflammatory symptoms but is not fully efficacious in all patients[49]. In cytokine release syndrome (CRS), as observed during CAR-T cell therapy or severe infections, rapid and sustained depletion of IL-6 may offer advantages over simple receptor antagonism, particularly in preventing downstream inflammatory cascades[50]. In RA, a heterogeneous autoimmune disease with both systemic and joint-specific

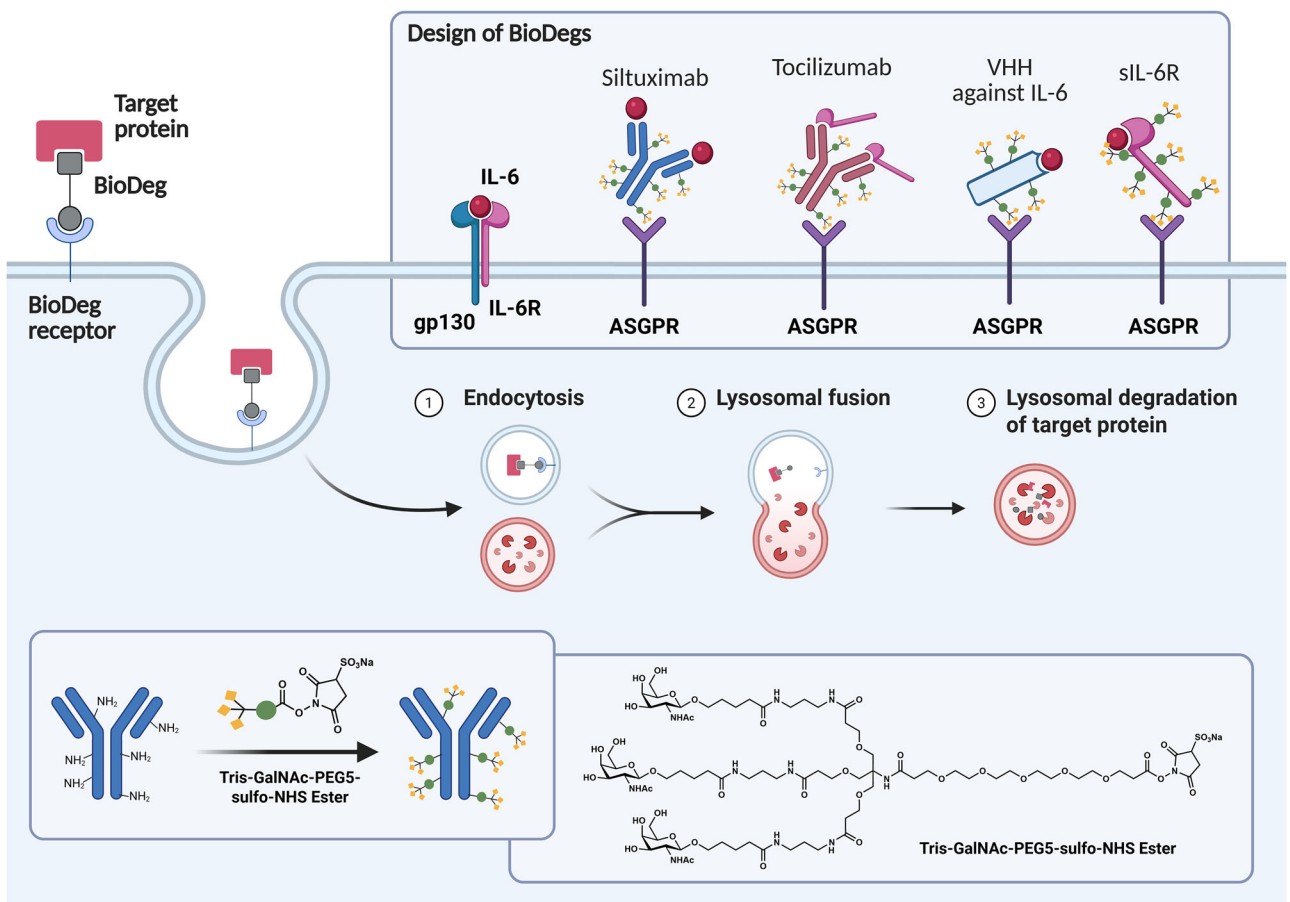

**Fig. 1 | Design and mechanism of glycan-based BioDeg for targeted lysosomal degradation.** Our BioDegs consist of a target-binding moiety (e.g., antibodies or VHH) conjugated to a TGN ligand that binds to asialoglycoprotein receptor (ASGPR) H1 on hepatocytes. Various target-binding molecules, including siltuximab, tocilizumab, VHH against IL-6, or soluble IL-6R (sIL-6R), were conjugated to TGN to induce ASGPR-mediated internalization. Upon binding, BioDeg:target complexes undergo (**1**) endocytosis, followed by (**2**) lysosomal trafficking and (**3**) degradation of the target protein in the lysosome. The TGN ligand used in this study is composed of the molecular structure Tris-GalNAc-PEG5-sulfo-NHS ester, enabling chemistry-specific conjugation via accessible primary amine groups.[90]

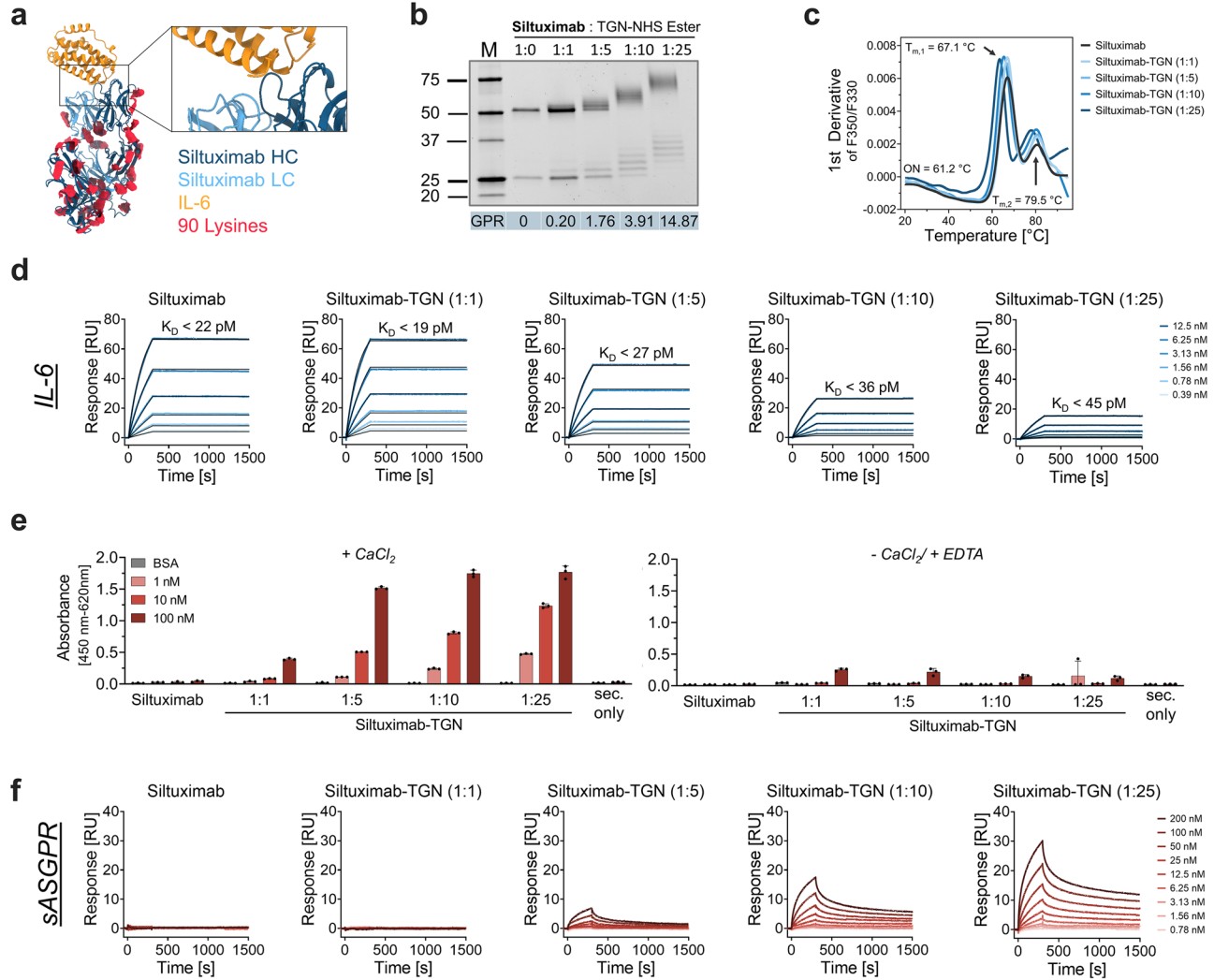

**Fig. 2 | Siltuximab-based BioDegs for targeting IL-6. a** AlphaFold-predicted complex structure of siltuximab (blue) and IL-6 (yellow). Positions of lysine residues available for TGN conjugation are shown (red). **b** SDS-PAGE under reducing conditions confirms successful, ratio-dependent TGN modification. GPR ratios derived from ESI-QTOF-MS are presented as mean values. **c** Thermal stability of siltuximab variants was assessed by nanoDSF. Data show unfolding behavior with respect to the degree of TGN labeling. Data represent the mean of three technical replicates. Parameters for unmodified siltuximab are shown. **d** Surface plasmon resonance (SPR) interaction kinetics of siltuximab and siltuximab-TGN variants with IL-6. IL-6 and sASGPR were immobilized on a CM5 sensor chip via amine coupling. Analytes were injected in a multi-cycle kinetic format at concentrations from 0.39 to 100 nM. Kinetic parameters ($K_D$, $k_{on}$, $k_{off}$) were obtained by global fitting using a 1:1 binding model. **e** ELISA-based binding assay of siltuximab(-TGN) to sASGPR. Detection was performed using anti-human-HRP, with absorbance measured at 450 nm (reference: 620 nm) after TMB development. Assays were conducted in HBS-N buffer supplemented with either 10 mM $CaCl_2$ or 10 mM EDTA to assess calcium dependence. A sec. only control was included to assess background signal from anti-human-HRP binding in the absence of sample incubation. Data are shown as mean ± SD of technical triplicates. **f** SPR-based binding analysis of siltuximab and siltuximab-TGN variants to sASGPR, performed at 25 °C in HBS-N buffer supplemented with 0.05% Tween-20 and 10 mM $CaCl_2$.

manifestations, blocking IL-6R can attenuate symptoms, yet does not fully reverse disease progression in many cases[51,52].

These observations support a growing consensus that mere neutralization of inflammatory cytokines may be insufficient in pathological settings characterized by high ligand concentrations or compensatory receptor expression. In such contexts, systemic clearance strategies may offer a more efficacious therapeutic solution by removing the effector molecule entirely from circulation. Leveraging ASGPR-mediated lysosomal trafficking to establish the liver as a clearance hub for circulating inflammatory mediators thus offers novel possibilities to therapeutically address inflammatory diseases.

In this study, we present a systematic experimental framework to evaluate BioDeg design variants and assess the feasibility of targeted protein degradation for modulating inflammatory cytokine signaling, using IL-6 and sIL-6R as clinically relevant model targets. For this purpose, we employ the well-established TGN modification strategy to engage ASGPR H1, with the objective of dissecting how degrader performance is shaped by combined effects of protein framework architecture and receptor engagement. To this end, we perform a head-to-head comparison of three distinct and widely used binder scaffolds: (i) full-length therapeutic antibodies (tocilizumab and siltuximab; DrugBank DB09036 and DB06273), representing high-affinity, clinically validated IgG frameworks; (ii) a compact camelid VHH selective for IL-6 (PMP6B629), exemplifying a small, rigid and high-affinity binder architecture; and (iii) a soluble variant of IL-6R (UniProt P08887) repurposed as a decoy-receptor degrader, representing a physiologically derived binding modality (Fig. 1; Supplementary Figs. 1–4). Our study design enables identification of scaffold-dependent advantages and limitations and lays out an experimental workflow for rational BioDeg development and optimization.

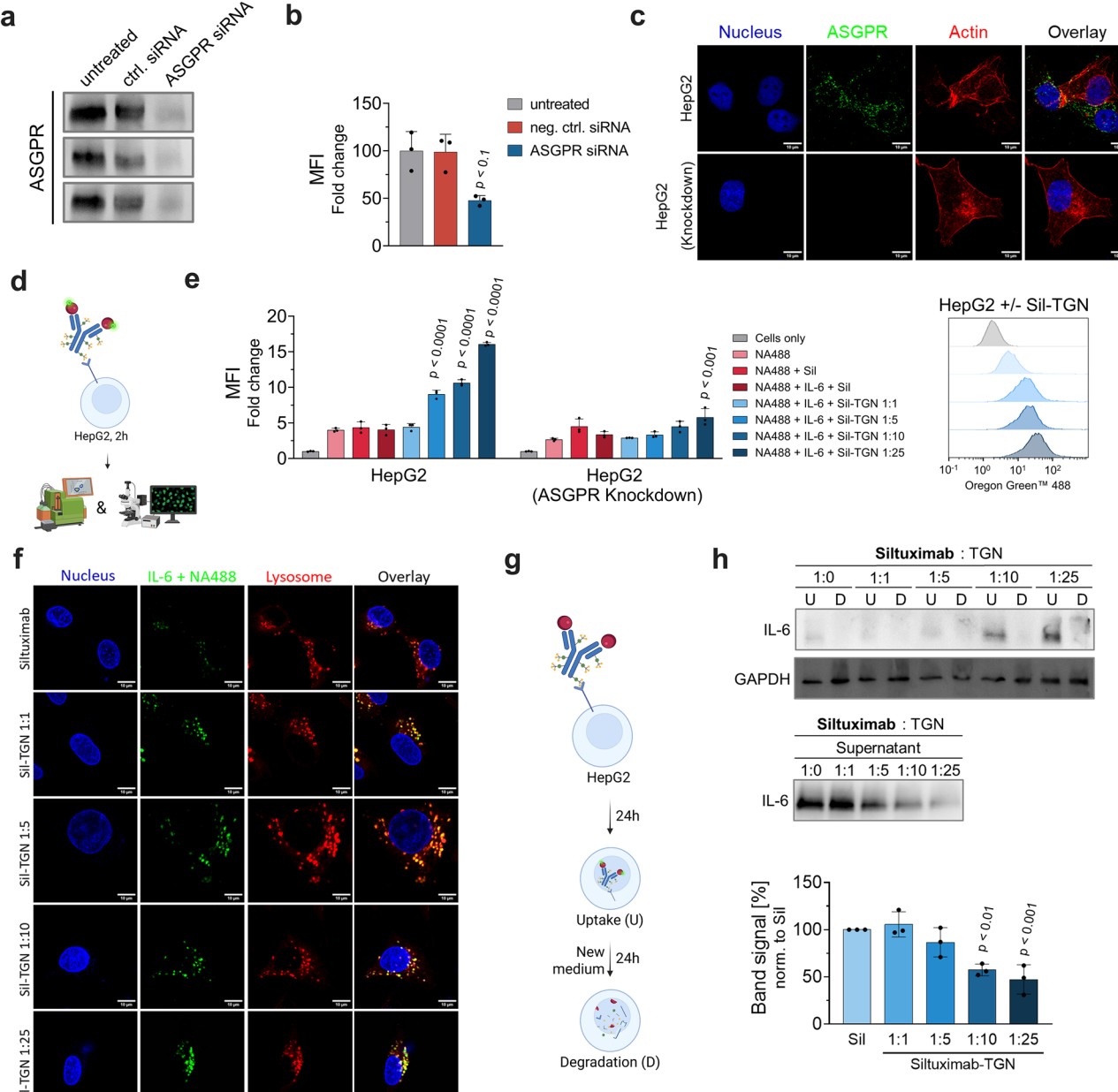

**Fig. 3 | Assessment of IL-6 degradation via siltuximab-TGN in HepG2 cells.**
**a** Immunoblot analysis of total ASGPR in untreated vs. ASGPR-siRNA-treated HepG2 cells. Non-targeting siRNA was used as control. Displayed are three biological replicates. **b** Flow cytometric surface staining of ASGPR on HepG2 cells. Data are normalized to fluorescence of untreated cells. **c** Immunofluorescence micrographs showing sub-cellular localization of ASGPR (green), nuclei (blue), and actin (red). Lack of ASGPR signal in knockdown HepG2 cell lines confirms successful gene silencing. **d** Schematic of target uptake assay design: serum-starved HepG2 cells are incubated for 2 h with NA488-labeled IL-6 in complex with siltuximab or respective TGN-labeled variants[91]. Uptake is analyzed by flow cytometry and live-cell confocal microscopy. **e** MFI fold change of IL-6 uptake normalized to untreated cells. **f** Live-cell confocal microscopy showing IL-6 (green), lysosomes (red; Lyso-Tracker™), and nuclei (blue; NucBlue™), confirming lysosomal localization. Scale bars = 10 μm. **g** Schematic of degradation assay[92]. BioDeg-target complexes were added to serum-starved HepG2 cells and incubated for 24 h. For analysis of uptake and depletion cell lysates ("U" - uptake) and supernatants were collected. In a parallel setup, medium was exchanged after 24 h of incubation and cultivation carried out for another 24 h. Lysates of this time point were collected to assess target clearance upon uptake ("D" - degradation). **h** Immunoblot analysis of lysates and supernatants assessing IL-6 degradation. Shown is a representative immunoblot for cell lysate samples and quantification of IL-6 in the supernatant samples. Statistical analysis: One-Way ANOVA (**b**, **h**)/ Two-Way ANOVA (**e**) analysis with Dunnet's multiple comparisons test was used. *P*-values indicate significant differences between cells treated with negative control (siltuximab + IL-6) and cells treated with samples. Data are mean ± SD of three biological replicates.

## Results

### Siltuximab-TGN: degradation of IL-6 via a monoclonal antibody

To obtain the chimeric (human-murine) anti-IL-6 siltuximab antibody we recombinantly produced heavy and light chain domains in HEK293T cells and purified the fully assembled antibody via Protein A affinity chromatography followed by a polishing step via size exclusion chromatography (Supplementary Fig. 1). Siltuximab contains a total of 90 lysines, of which 33 residues are found on each respective heavy chain and 12 on the respective light chains (Fig. 2a). For BioDeg generation via stoichiometric labeling we used triantennary β-D-GalNAc pentaethyleneglycol (PEG5) carboxylic acid N-hydroxysulfosuccinimide ester at molar ratios of 1:1, 1:5, 1:10, and 1:25. The chemical coupling resulted in concentration-dependent

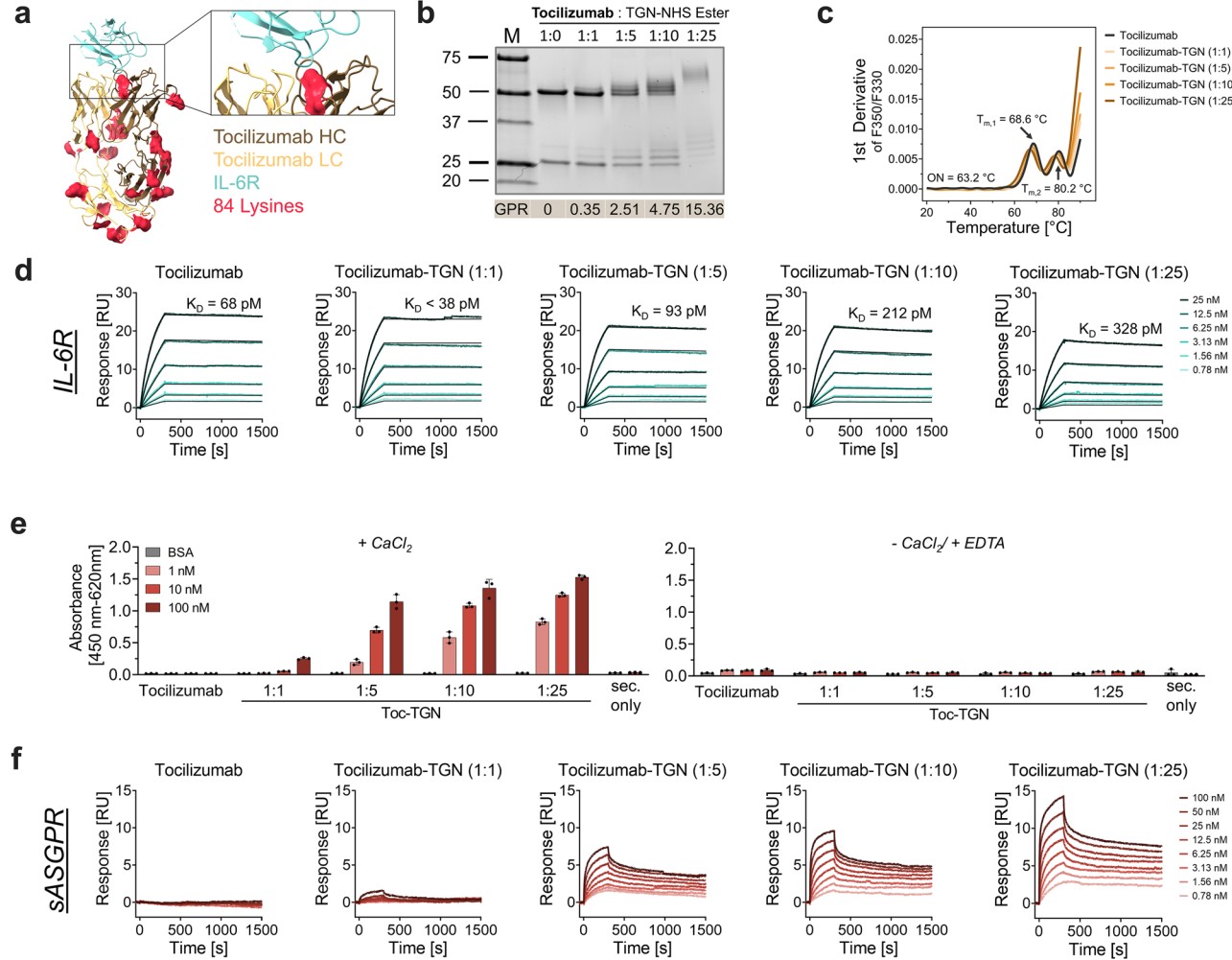

**Fig. 4 | Tocilizumab-based BioDegs for targeting sIL-6R. a** Cryo-EM structure of tocilizumab Fab (yellow) and IL-6R (blue) (PDB: 8J6F[93]). Positions of lysine residues for TGN conjugation are shown (red). **b** SDS-PAGE under reducing conditions confirms successful, ratio-dependent TGN modification. GPR ratios derived from ESI-QTOF-MS are presented as mean values. **c** Thermal stability of tocilizumab variants was assessed by nanoDSF. Data show unfolding behavior with respect to degree of TGN labeling. Data represent the mean of three technical replicates. Parameters for unmodified tocilizumab are shown. **d** Surface plasmon resonance (SPR) interaction kinetics of tocilizumab and tocilizumab-TGN variants with IL-6. IL-6 and sASGPR were immobilized on a CM5 sensor chip via amine coupling. Analytes were injected in a multi-cycle kinetic format at concentrations from 0.39 to 100 nM. Kinetic parameters ($K_D$, $k_{on}$, $k_{off}$) were obtained by global fitting using a 1:1 binding model. **e** ELISA-based binding assay of tocilizumab-(TGN) to sASGPR. Detection was performed using anti-human-HRP, with absorbance measured at 450 nm (reference: 620 nm) after TMB development. Assays were conducted in HBS-N buffer supplemented with either 10 mM $CaCl_2$ or 10 mM EDTA to assess calcium dependence. A sec. only control was included to assess background signal from anti-human-HRP binding in the absence of sample incubation. Data are shown as mean ± SD of technical triplicates. **f** SPR-based binding analysis of tocilizumab and tocilizumab-TGN variants to sASGPR, performed at 25 °C in HBS-N buffer supplemented with 0.05% Tween-20 and 10 mM $CaCl_2$.

shifts of molecular weight (Fig. 2b). To quantify glycan–protein ratios (GPRs), we employed electrospray ionization quadrupole time-of-flight mass spectrometry (ESI-QTOF-MS) (Supplementary Fig. 5a). Siltuximab exhibited sub-stoichiometric labeling at a 1:1 conjugation ratio (GPR = 0.2), whereas increasing the conjugation stoichiometry to 1:25 yielded a maximal GPR of 14.87 (Fig. 2b). Across all conditions, labeling of light-chain fragments was consistently less efficient than that of heavy-chain fragments. Following sample cleanup by ultrafiltration, we performed thermal stability analyses via nanoscale differential scanning fluorimetry (nanoDSF) to evaluate whether the decoration of the antibody with different amounts of TGN groups affected the integrity of siltuximab (Fig. 2c). Our data show a typical biphasic denaturation profile for all antibody variants, with minor differences in onset and melting temperatures after TGN labeling (siltuximab → siltuximab-TGN 1:25: Onset: 61.20 ± 0.35 °C → 57.49° ± 0.47 C; $T_{m,1}$: 67.05 ± 0.06 °C → 63.42 ± 0.13 °C; $T_{m,2}$: 79.46 ± 3.20 °C → 77.96 ± 0.13 °C) (Supplementary Tab. 1)[53]. Thus,

we concluded that the conditions applied to generate siltuximab-derived degraders did not affect the overall integrity of the IgG framework.

Next, we investigated whether the addition of PEG-linked TGN groups might have affected the binding of siltuximab to its antigen IL-6 (Supplementary Fig. 6). To determine the interaction kinetics of wildtype siltuximab and the glycan modified variants we performed surface plasmon resonance (SPR) measurements (Fig. 2d). We confirmed high-affinity IL-6 binding for all TGN-modified variants as well as wildtype siltuximab, with binding affinity ($K_D$) below 45 pM for all tested molecules. Despite maintaining high target-binding affinity upon TGN coupling, we observed slower association rates ($k_{on}$) [from $4.45 \times 10^5$ M$^{-1}$ s$^{-1}$ for wildtype to $2.21 \times 10^5$ M$^{-1}$ s$^{-1}$ for the 1:25 variant (Supplementary Tab. 2)], as well as decreases in RU$_{max}$ with increasing TGN modification levels, while dissociation remained extremely slow for all tested variants ($k_{off} < 10^{-5}$ s$^{-1}$, exceeding instruments limitations). In conclusion, SPR measurements show that TGN modification preserves high-affinity IL-6 binding, despite slightly slower association rates

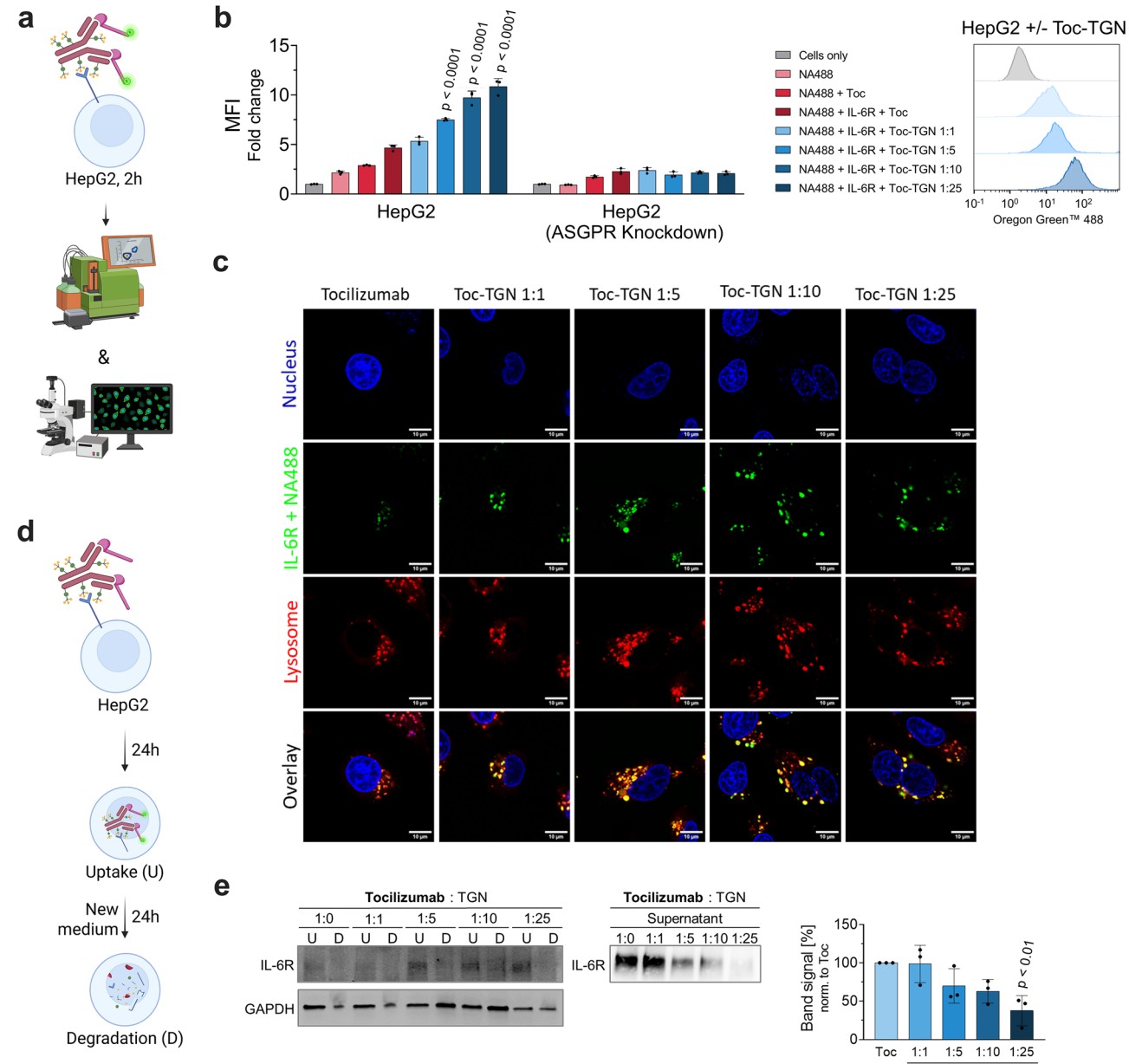

**Fig. 5 | Functional assessment of degradation of IL-6R via tocilizumab-TGN in HepG2 cells. a** Schematic of target uptake assay design: serum-starved HepG2 cells are incubated for 2 h with NA488-labeled IL-6 in complex with tocilizumab or respective TGN-labeled variants[91]. Uptake is analyzed by flow cytometry and live-cell confocal microscopy. **b** MFI fold change of IL-6R uptake normalized to untreated cells. **c** Live-cell confocal microscopy showing IL-6R (green), lysosomes (red; LysoTracker™), and nuclei (blue; NucBlue™), confirming lysosomal localization. Scale bars = 10 µm. **d** Schematic of degradation assay[92]. BioDeg-target complexes were added to serum-starved HepG2 cells and incubated for 24 h. For analysis of uptake and depletion cell lysates ("U" - uptake) and supernatants were collected.

In a parallel setup, medium was exchanged after 24 h of incubation and cultivation carried out for another 24 h. Lysates of this time point were collected to assess target clearance upon uptake ("D" - degradation). **e** Immunoblot analysis of lysates and supernatants assessing IL-6R degradation. Shown is a representative immunoblot for cell lysate samples and quantification of IL-6R in the supernatant samples. Statistical analysis: One-Way ANOVA (**e**)/ Two-Way ANOVA (**b**) analysis with Dunnet's multiple comparisons test was used. *P*-values indicate significant differences between cells treated with negative control (tocilizumab + IL-6R) and cells treated with samples. Data are mean ± SD of three biological replicates.

at higher modification levels, indicating that target engagement and potential biological performance are largely maintained.

In parallel, we established binding assays to verify the lectin-specific interaction between TGN-decorated siltuximab variants and the extracellular domain of the human asialoglycoprotein receptor H1 (here and in the following designated as sASGPR), which comprises the carbohydrate recognition domain. For this purpose, we immobilized sASGPR and evaluated glycan-binding properties via an indirect enzyme-linked immunosorbent assay (ELISA) (Fig. 2e). This setup revealed concentration-dependent binding of TGN-modified siltuximab variants at 1 nM, 10 nM, and 100 nM, whereas wild-type siltuximab showed no detectable binding at any of the tested concentrations. It should be noted that sASGPR is likely present as a monomer in solution, while plasma membrane ASGPR forms heterotrimers composed of ASGPR H1 and ASGPR H2[54]. Given that ASGPR is a calcium-dependent C-type lectin, we further tested binding specificity under calcium-depleted conditions using EDTA. In the absence of calcium, which acts as a critical prosthetic group for ligand binding, TGN-dependent interactions were almost completely abolished (Fig. 2e)[55]. These

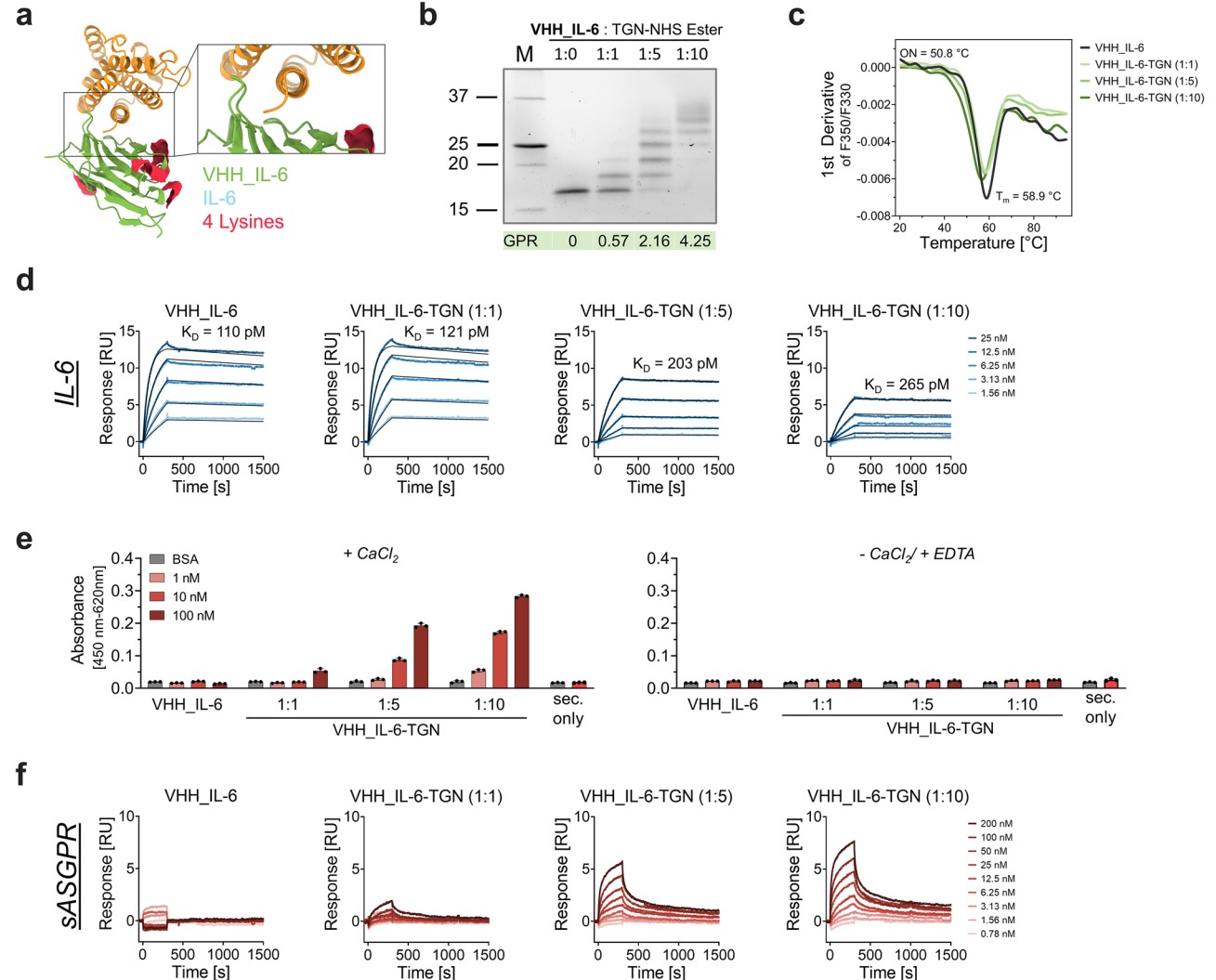

**Fig. 6 | VHH-based BioDegs for targeting IL-6. a** AlphaFold-predicted complex structure of VHH_IL-6 (green) and IL-6 (yellow). Positions of lysine residues for TGN conjugation are shown (red). **b** SDS-PAGE under reducing conditions confirms successful, ratio-dependent TGN modification. GPR ratios derived from ESI-QTOF-MS are presented as mean values. **c** Thermal stability of VHH_IL-6 variants were assessed by nanoDSF. Data show unfolding behavior with respect to the degree of TGN labeling. Data represent the mean of three technical replicates. Parameters for unmodified VHH_IL-6 are shown. **d** Surface plasmon resonance (SPR) interaction kinetics of VHH_IL-6 and VHH_IL-6-TGN variants with IL-6. IL-6 and sASGPR were immobilized on a CM5 sensor chip via amine coupling. Analytes were injected in a multi-cycle kinetic format at concentrations from 0.78 to 200 nM.

Kinetic parameters ($K_D$, $k_{on}$, $k_{off}$) were obtained by global fitting using a 1:1 binding model. **f** ELISA-based binding assay of VHH_IL-6(-TGN) to sASGPR. Detection was performed using anti-His-HRP, with absorbance measured at 450 nm (reference: 620 nm) after TMB development. Assays were conducted in HBS-N buffer supplemented with either 10 mM $CaCl_2$ or 10 mM EDTA to assess calcium dependence. Data are shown as mean ± SD of technical triplicates. A sec. only control was included to assess background signal from anti-His-HRP binding in the absence of sample incubation. **e** SPR-based binding analysis of VHH_IL-6 and VHH_IL-6-TGN variants to sASGPR, performed at 25 °C in HBS-N buffer supplemented with 0.05% Tween-20 and 10 mM $CaCl_2$.

results verify that upon TGN decoration siltuximab specifically interacts with ASGPR via its lectin property in a TGN conjugation stoichiometry dependent manner.

To investigate the impact of TGN avidity on ASGPR binding via an orthogonal method, we immobilized sASGPR on a SPR chip and measured association and dissociation responses of the respective siltuximab variants. Qualitative assessment of the SPR sensorgrams corroborated the results obtained from indirect ELISA, confirming specific binding of GalNAc-modified antibodies. Importantly, we observed a clear correlation between the degree of TGN labeling and the magnitude of the response units (RU), with higher labeling stoichiometries yielding increased RU signals (Fig. 2f). However, due to the multivalent nature of TGN-ASGPR interactions, accurate determination of kinetic parameters such as $k_{on}$, $k_{off}$, and the dissociation constant $K_D$ was not feasible using standard fitting models.

To investigate the relationship between antibody modification stoichiometries and the respective extent of target uptake and degradation, we employed HepG2 hepatocellular carcinoma cells as a model system and initially assessed IL-6 internalization by flow cytometry. To confirm receptor-specific uptake, we performed siRNA-mediated knockdown of *ASGR1* which encodes for ASGPR H1. Transfection with ASGPR-targeting siRNA resulted in a pronounced reduction of surface ASGPR protein levels, whereas control siRNA yielded no such effect (Fig. 3a, b). Flow cytometric surface staining revealed an approximate 52% ± 11% decrease in ASGPR signal upon knockdown, and western blot analysis of whole-cell lysates confirmed a substantial decrease in total ASGPR signal. Efficient knockdown was further confirmed by immunofluorescence staining, which revealed a complete loss of detectable ASGPR signal in HepG2 cells treated with *ASGR1*-targeting siRNA (Fig. 3c). Functional validation of receptor activity was demonstrated using a TGN-biotin/NeutrAvidin-488 reporter

system, revealing reduced NeutrAvidin-488 uptake in ASGPR-depleted cells as compared to untreated control HepG2 (Supplementary Fig. 7).

To minimize the influence of differential IL-6 binding kinetics among the various TGN-modified siltuximab variants, we pre-incubated each antibody with fluorescently labeled IL-6 (IL-6-biotin complexed with NeutrAvidin-488) prior to internalization (Fig. 3d). Our results reveal that IL-6 uptake increases in a TGN stoichiometry-dependent manner, with the 1:1 antibody-to-TGN ratio showing uptake comparable to unmodified siltuximab, while the 1:5, 1:10 and 1:25 variants exhibit significantly enhanced internalization (Fig. 3e). In *ASGPR*-silenced conditions uptake levels for siltuximab variants with labeling ratios of 1:1 to 1:10 did not show significant differences to negative control conditions. For the 1:25 variant we observed a slightly higher uptake than in control conditions, yet this increase was substantially lower than the effect observed in wild-type HepG2 cells (Fig. 3e). Live-cell confocal microscopy corroborated IL-6 internalization properties of siltuximab-based BioDegs, showing strong co-localization of IL-6 with the lysosome, dependent on the presence of TGN-labeled siltuximab (Fig. 3f). To evaluate the extent of overlap between lysosomal marker fluorescence and IL-6-specific fluorescence, we calculated Manders' coefficients (M1 and M2) based on our micrographs. We consistently observed M2 (fraction of IL-6 signal overlapping with lysosomal signal) values > 0.75 for all siltuximab-TGN variants, indicating high spatial co-localization and confirming efficient lysosomal trafficking of IL-6 (Supplementary Fig. 8).

Building on our internalization and subcellular localization studies, we next conducted target degradation assays to assess intracellular target accumulation, IL-6 clearance efficiencies, and depletion of extracellular IL-6 (Fig. 3g, h). Immunoblot analysis following 24 h of treatment with TGN-conjugated siltuximab in the presence of IL-6 revealed intracellular enrichment of IL-6 in a TGN avidity-dependent manner. Subsequent medium exchange and an additional 24 h of cultivation resulted in near-complete clearance of intracellular IL-6 for all tested degrader variants. In parallel, we tracked culture medium IL-6 levels after degrader treatment and observed a corresponding decrease that mirrored the patterns of intracellular accumulation. Densitometric analysis of IL-6 signals in cell supernatants demonstrate a TGN-ratio dependent reduction, with the 1:25 variant achieving 53% ± 16% depletion of extracellular IL-6 compared to unmodified siltuximab within 24 h of BioDeg activity (Fig. 3h).

## Tocilizumab-TGN: targeting the soluble IL-6 receptor

To further assess the applicability of glycan-based degraders for targeting soluble IL-6 receptor (sIL-6R), a key mediator of IL-6 trans-signaling, we selected the FDA-approved humanized monoclonal antibody tocilizumab as a further therapeutic IgG scaffold for TGN conjugation. Analogous TGN modification ratios were applied as previously established for our siltuximab-based degraders (Fig. 4a). SDS-PAGE analysis confirmed conjugation of light and heavy chain components. ESI-QTOF-MS analysis revealed a conjugation behavior for tocilizumab similar to that observed for siltuximab (Supplementary Fig. 5a, b). At a 1:1 Toc-TGN conjugation ratio, only sub-stoichiometric glycan labeling was detected (GPR = 0.35). Increasing the conjugation stoichiometry to 1:5 and 1:10 resulted in a progressive increase in glycan loading, yielding GPR values of 2.51 and 4.75, respectively, while the highest labeling ratio (1:25) produced a GPR of 15.36. NanoDSF measurements showed minor impact on thermal stability across the different variants (tocilizumab → tocilizumab -TGN 1:25: Onset: $63.23 \pm 0.08\ °C → 59.84 \pm 0.49\ °C$; $T_{m,1}$: $68.60 \pm 0.11\ °C → 67.00 \pm 0.03\ °C$; $T_{m,2}$: $80.24 \pm 0.23\ °C → 78.23 \pm 0.16\ °C$) (Fig. 4b, c, Supplementary Tab. 1).

Binding to sIL-6R was confirmed for all glycosylated tocilizumab variants with a slight decrease of signal observed with increasing glycan abundance (Fig. 4d, Supplementary Fig. 6). In contrast to siltuximab, the kinetic profiles for target-binding exhibited stronger TGN modification dependent differences resulting in up to 5-fold weaker affinity as compared to wildtype tocilizumab. This trend was attributed to both higher $k_{off}$ rates and slightly lower $k_{on}$ rates with increasing degrees of conjugation. Interaction between TGN-decorated tocilizumab variants and sASGPR was

confirmed in calcium- and labeling degree-dependent manner (Fig. 4e, f; Supplementary Tab. 2).

Our ASGPR-selective variants of tocilizumab enabled targeted removal of sIL-6R in HepG2 cells (Fig. 5). Flow cytometric analysis showed that cellular uptake of sIL-6R-NA488 increased progressively with the extent of TGN conjugation (Fig. 5a, b). In ASGPR-knockdown cells, this uptake was abrogated, confirming that internalization was ASGPR-dependent (Fig. 5b). Microscopic co-localization studies support these findings revealing overlap of sIL-6R and lysosomal signals upon treatment with tocilizumab-TGN variants, with Manders' coefficients of M2 > 0.79 (Supplementary Fig. 8). These values indicate effective intracellular routing of sIL-6R to degradative compartments via the endo-lysosomal pathway (Fig. 5c). Degradation studies provided further evidence for sIL-6R clearance (Fig. 5d, e). After 24 h of treatment with tocilizumab-TGN, sIL-6R accumulated within cells, and was mostly eliminated within 24-h. We further detected a TGN-dose-dependent decrease in extracellular sIL-6R levels, with the 1:25 variant achieving a reduction of approximately 62% ± 20% compared to the unmodified antibody.

## Anti-IL-6 VHH-TGN: a camelid immunoglobulin-based BioDeg targeting IL-6

Having established the feasibility of targeting both IL-6 and soluble IL-6 receptor (sIL-6R) using TGN-based BioDegs with full-length antibody scaffolds, we next explored whether a significantly smaller molecular framework, such as a variable heavy domain fragment (VHH) derived from a camelid heavy-chain antibody, could achieve similar functional outcomes. VHH fragments were selected due to their high target-binding affinities, ease of production, small size, and potential for rapid tissue penetration, despite containing substantially fewer lysine residues. To this end, we selected a previously published anti-IL-6 VHH and recombinantly produced it using a bacterial expression system and purified it via affinity chromatography (Supplementary Fig. 3)[56]. Given the substantially lower number of only 4 reactive lysine residues per VHH as compared to whole IgGs (Fig. 6a), we selected different TGN coupling stoichiometries as before, with the highest VHH to TGN ratio being 1:10. Notably, we observed a laddering pattern of distinct VHH bands with the abundance of higher molecular weight bands increasing in a labeling-degree dependent manner (Fig. 6b). This trend resembles the signals we observed for antibody light chain labeling, while heavy chain modifications of both siltuximab and tocilizumab resulted in rather diffuse signals, suggesting a more heterogeneous TGN distribution for these larger molecules that exhibit a higher number of primary amine groups. ESI-QTOF-MS analysis of our VHH constructs showed a conjugation profile comparable to that observed for full-length antibodies, despite the lower overall number of available lysine residues (Supplementary Fig. 5c). At a 1:1 conjugation ratio, a GPR of 0.57 was obtained, which increased to 2.16 and 4.25 upon raising the stoichiometry to 1:5 and 1:10, respectively. Thermal denaturation profiles of the respective VHH-TGN variants did not indicate considerable instability upon the addition of glycan groups at any of the selected labeling stoichiometries (Fig. 6c, Supplementary Tab. 1).

IL-6 binding analyses showed that the unmodified VHH bound selectively and with high affinity, exhibiting a $K_D$ of 110 pM (Fig. 6d, Supplementary Fig. 6, Supplementary Tab. 2). Conjugation of TGN at a 1:1 molar ratio resulted in comparable association and dissociation kinetics to unmodified VHH, while the variants modified at 1:5 and 1:10 ratios displayed approximately twofold weaker affinities (Fig. 6d). In accordance to the limited number of modifiable lysine residues, we observed considerably weak sASGPR binding signals by both ELISA and SPR, with only the two higher TGN-modification degrees displaying dose- and calcium- dependent binding well detectable above noise thresholds (Fig. 6e, f).

Despite its small size (15.7 kDa) and lower TGN avidity, our VHH-BioDeg variant demonstrated efficient cellular uptake and target degradation. Flow cytometric uptake experiments confirmed TGN-dependent internalization of IL-6, with the variants modified by coupling ratios of 1:1 and 1:5 having significant but rather small effect on IL-6 uptake as compared

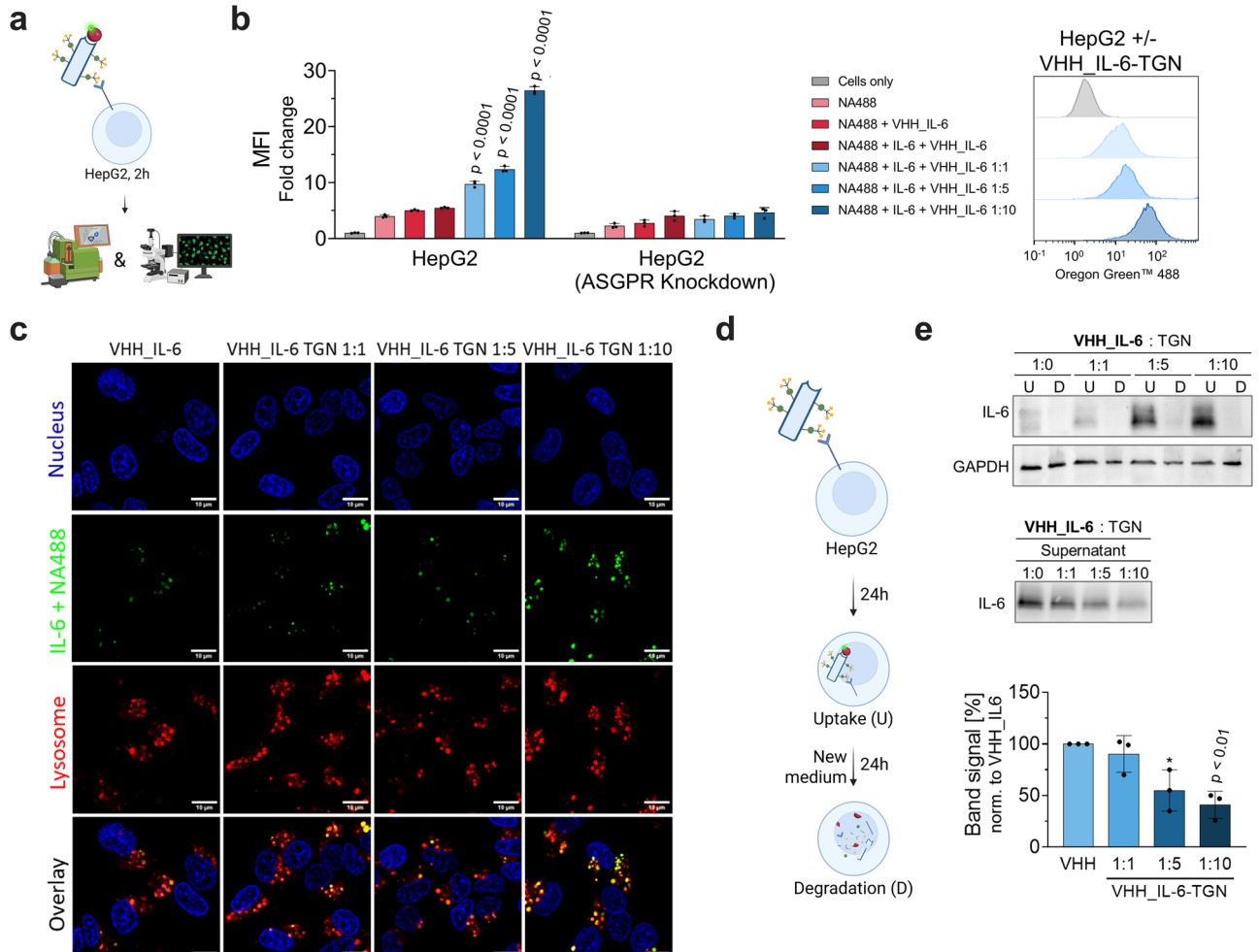

**Fig. 7 | Functional assessment of degradation of IL-6 via VHH_IL-6-TGN in HepG2 cells. a** Schematic of target uptake assay design: serum-starved HepG2 cells are incubated for 2 h with NA488-labeled IL-6 in complex with VHH_IL-6 or respective TGN-labeled variants[91]. Uptake is analyzed by flow cytometry and live-cell confocal microscopy. **b** MFI fold change of IL-6 uptake normalized to untreated cells. **c** Live-cell confocal microscopy showing IL-6 (green), lysosomes (red; Lyso-Tracker™), and nuclei (blue; NucBlue™), confirming lysosomal localization. Scale bars = 10 μm. **d** Schematic of degradation assay[92]. BioDeg-target complexes were added to serum-starved HepG2 cells and incubated for 24 h. For analysis of uptake and depletion cell lysates ("U" - uptake) and supernatants were collected. In a

parallel setup, medium was exchanged after 24 h of incubation and cultivation carried out for another 24 h. Lysates of this time point were collected to assess target clearance upon uptake ("D" - degradation). **e** Immunoblot analysis of lysates and supernatants assessing IL-6 degradation. Shown is a representative immunoblot for cell lysate samples and quantification of IL-6 in the supernatant samples. Statistical analysis: One-Way ANOVA (**e**)/ Two-Way ANOVA (**b**) analysis with Dunnet's multiple comparisons test was used. *P*-values indicate significant differences between cells treated with negative control (VHH_IL-6 + IL-6) and cells treated with samples. Data are mean ± SD of three biological replicates.

to VHH with the highest degree of modification (Fig. 7a, b). Live-cell confocal microscopy demonstrated lysosomal co-localization of IL-6 with the strongest signal intensity for IL-6 observed with VHH-TGN 1:10, while unmodified VHH and TGN ratios 1:1 and 1:5 showed weak uptake signals presumably attributable to ASGPR-independent mechanisms (Fig. 7c).

Degradation assays and immunoblot readouts, using our methodology outlined for the antibody degrader variants above, showed intracellular IL-6 accumulation after 24 h incubation with VHH degraders, followed by target clearance upon medium exchange. Quantification of IL-6 in the supernatant demonstrated a TGN-dose-dependent decrease, with the 1:10 variant reducing IL-6 to 41% ± 13% of unmodified VHH control levels (Fig. 7d,e).

### sIL-6R-TGN: decoy receptor-based targeting of IL-6

After establishing the feasibility of using immunoglobulin scaffolds for cytokine degradation, we next investigated whether alternative IL-6-binding molecules could function as effective biological degraders (Fig. 8a). To this end, we repurposed sIL-6R as a decoy receptor for TGN-mediated

lysosomal targeting via ASGPR. We generated four TGN-conjugated sIL-6R variants with molar coupling ratios of 1:1, 1:5, 1:10, and 1:25. SDS-PAGE analysis revealed conjugation-dependent mass shifts toward higher molecular weights, indicating successful concentration-dependent glycan modification (Fig. 8b). Considering the limited number of 12 theoretically modifiable lysine residues of sIL-6R, we infer that the 1:25 conjugation ratio likely results in near-complete lysine occupancy, which is supported by the lack of lower molecular weight bands at the level of unmodified sIL-6R upon conjugation. The determination of accurate GPR values for sIL-6R conjugates was inherently limited by the strong and highly heterogeneous glycosylation of the receptor. The broad distribution of endogenous glycoforms resulted in extensive peak overlap and signal broadening in ESI-QTOF-MS spectra, which impeded reliable deconvolution and precise assignment of mass shifts attributable to glycan conjugation. As a consequence, no GPR values could be determined for sIL-6R conjugates. Biophysical analysis via nanoDSF showed a thermal unfolding onset at 44 °C and a melting temperature ($T_m$) of approximately 51 °C, with no significant differences across variants (Fig. 8c, Supplementary Tab. 1). These results

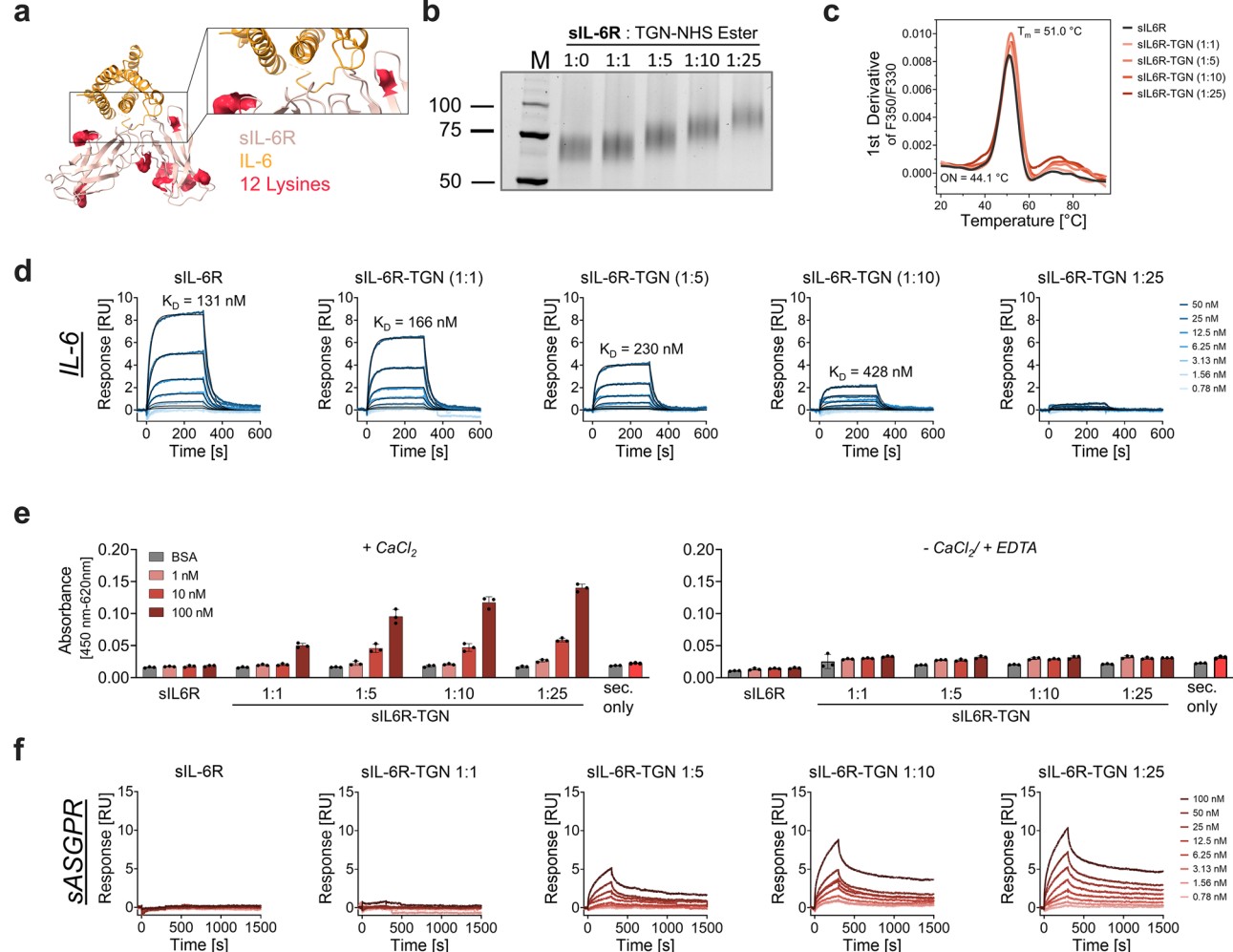

**Fig. 8 | sIL-6R-based BioDegs for targeting IL-6. a** Crystal structure of of sIL-6R (rose) and IL-6 (yellow) (PDB: 1P9M[94]). Positions of lysine residues for TGN conjugation are shown (red). **b** SDS-PAGE under reducing conditions confirms successful, ratio-dependent TGN modification. **c** Thermal stability of sIL-6R variants was assessed by nanoDSF. Data show unfolding with respect to the degree of TGN labeling. Data represent the mean of three technical replicates. Parameters for unmodified sIL-6R are shown. **d** Surface plasmon resonance (SPR) interaction kinetics of sIL-6R and sIL-6R-TGN variants with IL-6. IL-6 and sASGPR were immobilized on a CM5 sensor chip via amine coupling. Analytes were injected in a multi-cycle kinetic format at concentrations from 0.39 to 100 nM. Kinetic parameters ($K_D$, $k_{on}$, $k_{off}$) were obtained by global fitting using a 1:1 binding model.

**f** ELISA-based binding assay of sIL-6R(-TGN) to sASGPR. Detection was performed using anti-His-HRP, with absorbance measured at 450 nm (reference: 620 nm) after TMB development. Assays were conducted in HBS-N buffer supplemented with either 10 mM CaCl$_2$ or 10 mM EDTA to assess calcium dependence. Data are shown as mean ± SD of technical triplicates. A sec. only control was included to assess background signal from anti-human-HRP binding in the absence of sample incubation. (**e**) SPR-based binding analysis of sIL-6R and sIL-6R-TGN variants to sASGPR, performed at 25 °C in HBS-N buffer supplemented with 0.05% Tween-20 and 10 mM CaCl$_2$.

demonstrate that even at high modification levels, TGN conjugation does not compromise the structural integrity of sIL-6R.

Characterization of binding kinetics revealed that wild-type sIL-6R exhibits both a rapid association ($k_{on} = 2.48 \times 10^5$ M$^{-1}$ s$^{-1}$) and a fast dissociation rate ($k_{off} = 0.035$ s$^{-1}$), resulting in a binding affinity of $K_D = 131$ nM (Fig. 8d; Supplementary Tab. 2). This affinity is more than three orders of magnitude weaker than observed for siltuximab and the anti-IL-6 VHH under comparable conditions in this study. Furthermore, TGN conjugation markedly affected the kinetic profile of sIL-6R. At a labeling stoichiometry of 1:5, the apparent affinity decreased to 428 nM, and higher degrees of modification led to a progressive decrease in binding response (Fig. 8d, Supplementary Fig. 6). The most abundantly labeled variant (1:25) showed no detectable binding in both ELISA and SPR measurements. Our data suggest that extensive glycan modification likely introduces steric hindrance and potentially blocks lysine residues in proximity to the IL-6 binding interface (K152 and K271), thereby impairing ligand interaction. In contrast to this trend, sASGPR binding

increased in a TGN avidity-dependent manner, as confirmed by SPR and ELISA (Fig. 8e, f).

Flow cytometry analysis in HepG2 cells revealed that unliganded sIL-6R-TGN was efficiently internalized in a concentration-dependent manner (Fig. 9a–c). Higher TGN-to-protein ratios correlated with greater internalization of the degrader molecule itself. Importantly, *ASGPR* knockdown via siRNA significantly reduced cellular uptake across all TGN variants, confirming that internalization was specifically mediated via the ASGPR pathway. This demonstrated that sIL-6R is a suitable carrier for glycan-based receptor targeting.

However, in contrast to internalization of ligand-free sIL-6R, pre-assembled complexes of sIL-6R-TGN and IL-6 resulted in only minor uptake of IL-6 (Fig. 10a–c). Flow cytometry and western blot analyses revealed that IL-6 levels in supernatants and lysates remained largely unchanged after treatment with sIL-6R-TGN-IL-6 complexes. A slight reduction in IL-6 levels was observed at higher TGN ratios, but this effect was only marginal compared to controls, and substantially lower than the

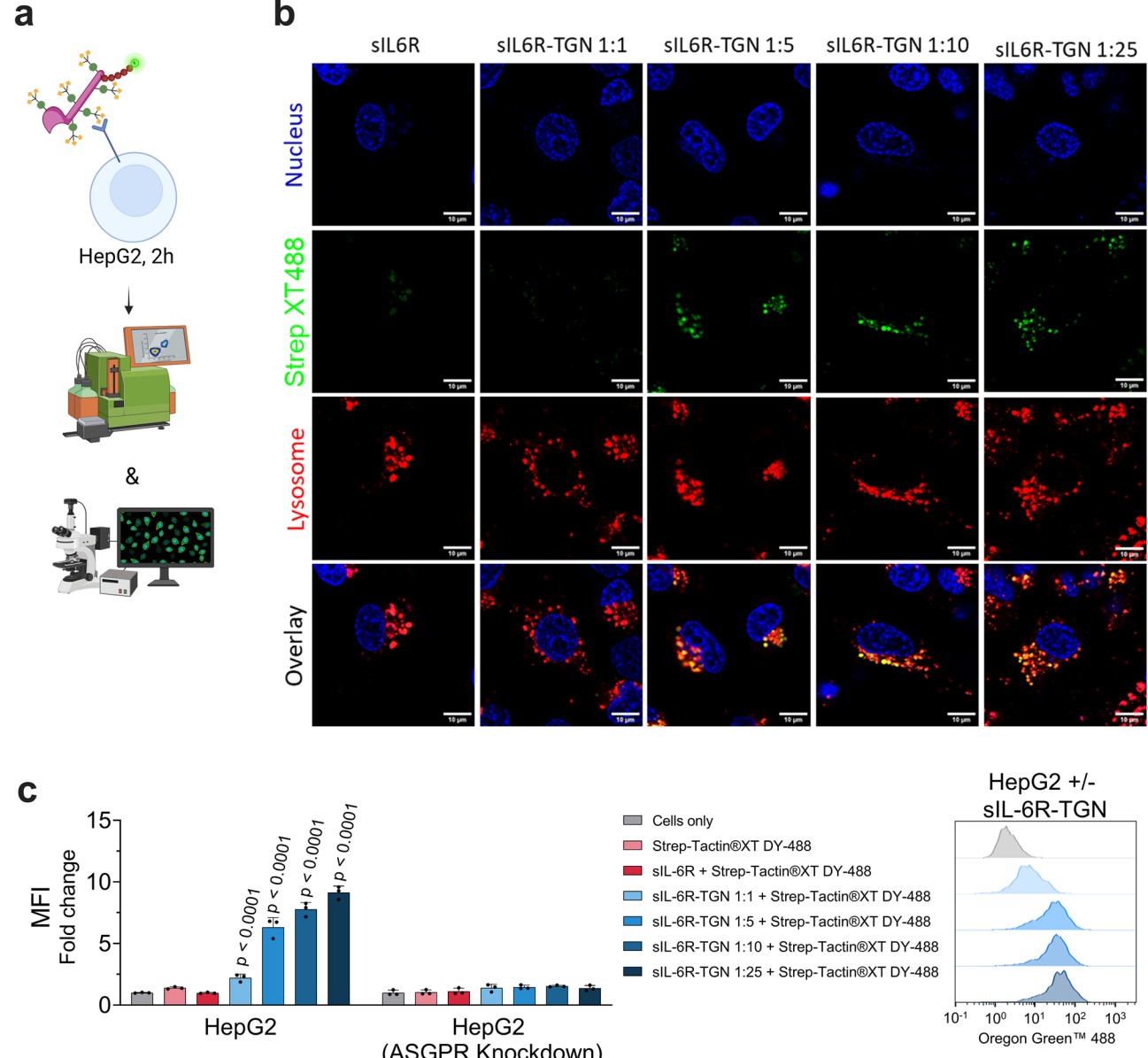

**Fig. 9 | Functional assessment of uptake of sIL-6R-TGN in HepG2 cells.**
**a** Schematic of target uptake assay design: serum-starved HepG2 cells are incubated for 2 h with StrepTactin XT DY488-labeled IL-6 in complex with sIL-6R or respective TGN-labeled variants[91]. Uptake is analyzed by flow cytometry and live-cell confocal microscopy. **b** Live-cell confocal microscopy showing sIL-6R(-TGN) (green), lysosomes (red; LysoTracker™), and nuclei (blue; NucBlue™), confirming lysosomal localization. Scale bars = 10 μm. **c** MFI fold change of IL-6 uptake normalized to untreated cells. Statistical analysis: Two-Way ANOVA analysis with Dunnet's multiple comparisons test was used. *P*-values indicate significant differences between cells treated with negative control (sIL-6R) and cells treated with samples. Data are mean ± SD of three biological replicates.

degradation observed with siltuximab-TGN or VHH_IL-6-TGN (Fig. 10d, e). This suggests that the IL-6/sIL-6R complex may not be stable when diluted in growth medium, likely caused by rapid off-rate kinetics. Confocal microscopy supported these findings, showing that sIL-6R-TGN accumulated within lysosomes, whereas the IL-6 signal appeared weak and largely nonspecific, with no differential effects in fluorescence intensity or intracellular abundance across the various sIL-6R variants. We infer that these non-differential signals could originate from free IL-6 molecules that interact with HepG2 cells through an ASGPR independent mechanism, presumably via the low-abundant native IL-6Rα on the plasma membrane. Our data provide evidence that sIL-6R-TGN variants are efficiently internalized in an ASGPR-mediated, and TGN-avidity-dependent manner by HepG2 cells. However, their

capacity to act as efficacious degraders for IL-6 was inferior to that of antibody- or VHH-based BioDegs. The limited IL-6 clearance efficiency underscores the importance of modulating target binding and receptor engagement kinetics for efficient BioDeg function.

In order to establish head-to-head assessments of target uptake efficiencies, the best-performing BioDeg variants with respect to IL-6 internalization and clearance from each framework design were evaluated in a time-course uptake assay (Fig. 10f). Within our assay window, we observed that at any given time point the VHH-based degrader showed highest IL-6-uptake, followed by siltuximab-TGN. After 24 h the VHH-degrader induced 47% ± 5% higher internalization as compared to our IgG-based degrader. Notably, while the decoy receptor molecule did also cause a time-dependent increase in IL-6 fluorescence, we determined that this

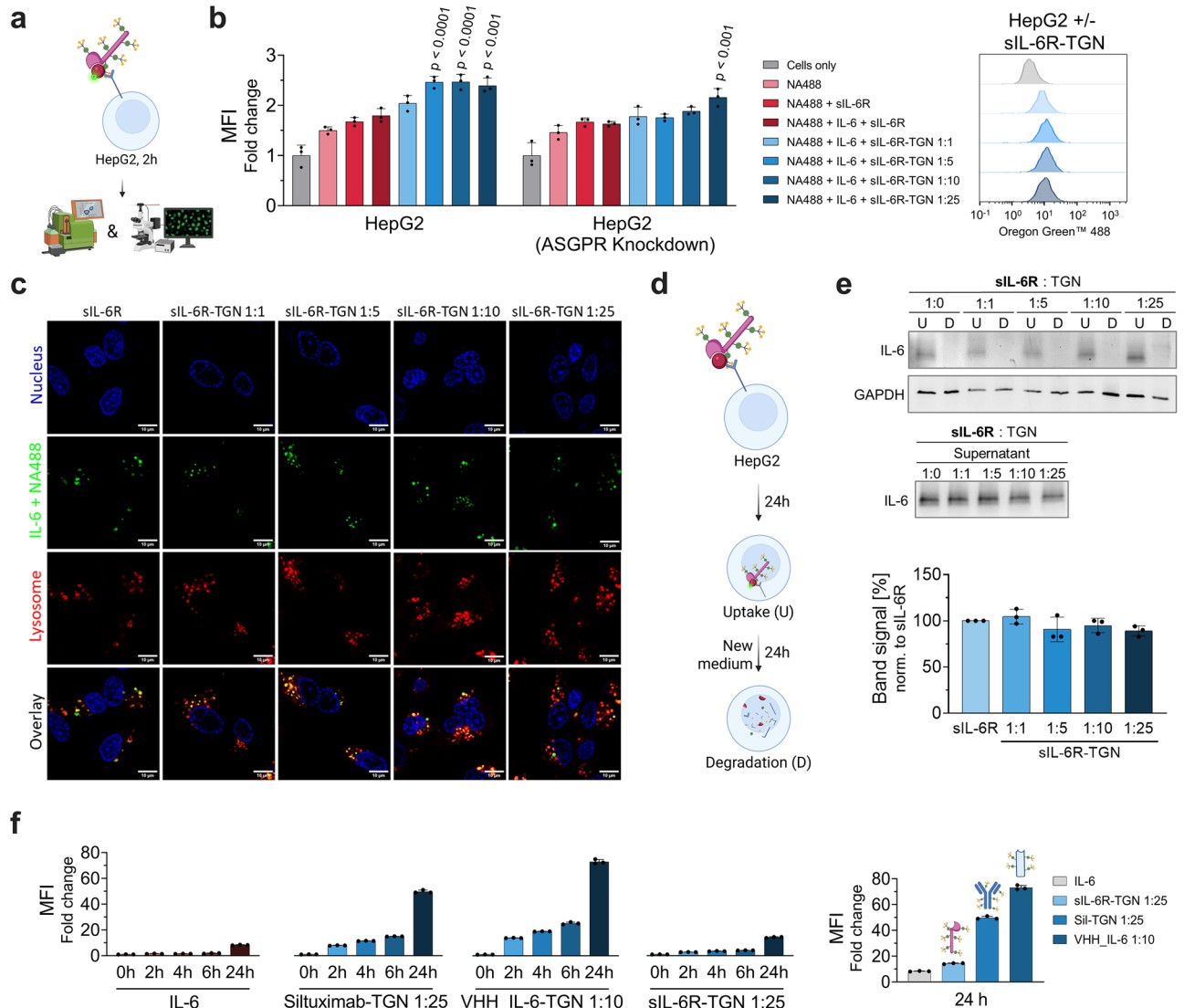

**Fig. 10 | Functional assessment of degradation of IL-6 via sIL-6R-TGN in HepG2 cells. a** Schematic of target uptake assay design: Serum-starved HepG2 cells are incubated for 2 h with NA488-labeled IL-6 in complex with sIL-6R or respective TGN-labeled variants. Uptake is analyzed by flow cytometry and live-cell confocal microscopy[91]. **b** MFI fold change of IL-6 uptake normalized to untreated cells. **c** Live-cell confocal microscopy showing IL-6 (green), lysosomes (red; LysoTracker™), and nuclei (blue; NucBlue™), confirming lysosomal localization. Scale bars = 10 μm. **d** Schematic of degradation assay[92]. BioDeg-target complexes were added to serum-starved HepG2 cells and incubated for 24 h. For analysis of uptake and depletion cell lysates ("U" - uptake) and supernatants were collected. In a parallel setup, medium was exchanged after 24 h of incubation and cultivation carried out for another 24 h. Lysates of this time point were collected to assess target clearance upon uptake ("D" - degradation). **e** Immunoblot analysis of lysates and supernatants assessing IL-6 degradation. Shown is a representative immunoblot for cell lysate samples and quantification of IL-6 in the supernatant samples. **f** Time-course of IL-6 uptake (2 h, 4 h, 6 h and 24 h) measured by flow cytometry for IL-6 alone and in complex with siltuximab-TGN (1:25), VHH_IL-6-TGN (1:10), and sIL-6R-TGN (1:25). Statistical analysis: One-Way ANOVA/ Two-Way ANOVA analysis with Dunnet's multiple comparisons test was used. *P*-values indicate significant differences between cells treated with negative control (sIL-6R + IL-6) and cells treated with samples. Data are mean ± SD of three biological replicates.

internalization was only marginally higher than the uptake of free IL-6 by HepG2.

## Discussion

The advent of targeted protein degradation (TPD) has introduced a transformative shift in therapeutic strategies by enabling the active removal of disease-relevant proteins rather than their inhibition or neutralization[5]. While proteasome-based approaches such as PROTACs and molecular glues have successfully expanded the druggable proteome to previously intractable intracellular targets, their mechanism of action is inherently limited to the cytoplasmic and nuclear compartments[57,58]. This has led to the emergence of lysosomal targeting degraders, engineered protein-based constructs capable of mediating the endocytosis and subsequent lysosomal

degradation of extracellular or membrane-associated proteins, thereby significantly expanding the therapeutic scope of TPD[59].

By harnessing lysosomal trafficking pathways, biological degraders enable the selective clearance of membrane-located or extracellular disease drivers, such as soluble antigens, circulating cytokines, auto-antibodies and membrane-bound receptors, that are typically inaccessible to intracellular degradation machineries. This approach is particularly valuable in pathological contexts characterized by persistent or excessive antigen exposure, where conventional therapies often fall short[60].

Conditions such as chronic inflammation, autoimmunity, allergy, or cancer frequently involve sustained antigen presence that fuels disease progression, making targeted degradation a promising strategy to intervene in antigen-driven pathology[61-65]. In the context of oncology, LYTACs have

been designed to degrade membrane-bound targets that involved in immune-checkpoint processes. For example, Banik et al. demonstrated that LYTACs directed against PD-L1 via the CI-M6PR pathway led to internalization and lysosomal degradation of this checkpoint protein[15]. Similarly, TGN-based LYTACs targeting EGFR via ASGPR have effectively attenuated receptor signaling and led to specific target protein degradation in cancer cell models[14]. These findings highlight how biological degraders can modulate the tumor microenvironment and anti-tumor response by removing essential surface proteins, offering a new dimension in precision oncology. In parallel to oncology, biological degraders are gaining traction in chronic immune and allergic diseases. One of the most promising candidates emerging from Lycia Therapeutics' pipeline is LCA-0061, a CataLYTAC-based degrader specifically designed to target immunoglobulin E (IgE). In preclinical studies involving non-human primates, a single administration resulted in near-complete and sustained depletion of both free and total IgE[66]. This performance surpasses that of omalizumab, which neutralizes free IgE but has minimal impact on overall IgE levels[67,68].

In this study, we set out to introduce a systematic approach for validation of different TGN-based BioDeg design variants, ensuring robust assessment of target engagement, cellular uptake, lysosomal trafficking, and functional degradation within a unified experimental framework (Supplementary Fig. 13).

Overall, our data underscore the critical importance of scaffold architecture and affinity tuning for optimization of functional performance of glycan-based BioDegs. Specifically, we observed marked differences in degradation efficiency between variants that differed in target-binding domains (antibody, VHH, soluble receptor) and valency of ASGPR-targeting moieties. The full-length antibody-based degraders, while displaying very high target affinity in the low picomolar range, showed moderate degradation efficacy. The VHH-based degrader, in contrast, achieved superior degradation despite exhibiting lower affinity in the mid-picomolar range and carrying fewer TGN motifs. The soluble IL-6 receptor (sIL-6R), with a target affinity in the high nanomolar range, was the least effective in facilitating IL-6 degradation. Importantly, TGN modification of the antibody- and VHH-based constructs did not interfere with their neutralizing properties in an IL-6 reporter assay, confirming preserved biological activity upon conjugation (Supplementary Fig. 9). In contrast, the sIL-6R-based BioDeg showed no measurable inhibition of IL-6 signaling in a reporter assay, which is consistent with its physiological role as a co-receptor rather than a direct IL-6 antagonist. It is noteworthy that although neutralization of a target antigen is not a mechanistic requirement for biological degraders, it may nonetheless confer additional advantages in certain contexts. Simultaneous neutralization and degradation could act synergistically to enhance silencing of pathogenic immune pathways. Moreover, neutralizing engagement may help limit unintended degradation of interacting proteins and reducing the likelihood of co-internalization of undesired ternary complex partners that associate with the protein of interest in distinct physiological settings. In cytokine biology, this consideration becomes relevant if a non-neutralizing IL-6 binder were to promote degradation of IL-6–sIL-6R–sgp130 complexes, potentially reducing the neutralizing capacity of sgp130. As sgp130 acts as a shared trans-signaling inhibitor for both IL-6 and IL-11, such an effect could alter the balance of gp130-dependent cytokine signaling[69]. Hence, a deeper understanding about biochemical pathways and context dependent PPIs is essential to guide the targeting of the best suited interaction surfaces.

While target-binding affinity and ASGPR engagement cannot be fully decoupled due to scaffold-dependent differences in TGN valency and presentation, our comparative analysis enables identification of the most effective BioDeg architectures under realistic design constraints. Our findings reveal a non-linear multifaceted relationship between target affinity, TGN valency, and functional outcome. One explanation for the superior performance of the VHH-based construct may lie in its small molecular size (~15 kDa), which could facilitate more efficient access to target epitopes and improved ternary complex formation. Furthermore, the rigid architecture of the VHH construct may allow for better spatial alignment between the target

and ASGPR, minimizing steric hindrance and inter-domain flexibility which could originate from the multi-subunit composition of whole IgGs. The kinetics of ASGPR-specific clathrin-dependent endocytosis have been discussed previously with data suggesting differential uptake efficiencies depending on the size and type of cargo[70]. Correspondingly, our data demonstrate that a smaller molecular framework with similar target-binding kinetics and lower TGN avidity is internalized at higher rates than its IgG counterpart with superior target and receptor-binding. Our live cell imaging data demonstrate efficient lysosomal co-localization of all BioDeg constructs. Looking forward, examining early versus late endosomal markers could provide additional insights into trafficking kinetics and sorting efficiency. In the case of our decoy-degrader, in vitro data demonstrate efficient internalization of sIL-6R, but only minimal uptake of IL-6. Given the pronounced biochemical differences among the various sIL-6R conjugates, three key factors likely contribute to the observed disconnect between BioDeg uptake and target internalization.

First, the kinetic profile of unmodified sIL-6R for IL-6 binding is characterized by a rapid dissociation rate and an overall affinity that is three orders of magnitude weaker than that of the VHH and antibody variants tested under identical experimental conditions. Second, we observe a TGN stoichiometry–dependent decline in IL-6 binding affinity due to steric interference, which is expected to shift the binding equilibrium toward free IL-6 at 100 nM concentrations of IL-6 and sIL-6R-TGN in the culture medium. These unfavorable kinetic properties are consistent with our cellular data, where unliganded sIL-6R-based degraders are efficiently internalized, yet IL-6 uptake is minimal across all tested conditions (Figs. 9, 10). Given these observations, it seems presumable that the high affinity of TGN for ASGPR promotes preferential internalization of decoy degraders, while IL-6 remains in the extracellular space. Mamidyala et al. reported a dissociation constant ($K_D$) of 40.4 ± 9.5 μM for monomeric GalNAc, reflecting its relatively low affinity for ASGPR[71]. In contrast, Biessen et al. described significantly enhanced binding for trivalent GalNAc (TGN) constructs, reporting a $K_D$ of approximately 200 nM for a TGN derivative with a 20 Å linker[72]. This multivalency effect substantially improves receptor engagement and is consistent with the efficient uptake of TGN-modified constructs[73]. In a related approach, Sanhueza and colleagues have reported alternative ASGPR ligands based on multivalent bicyclic bridged ketals, which exhibit stronger receptor binding than GalNAc motifs and enable selective hepatocyte accumulation in rodent models[74].

Recent work further highlights alternative strategies for ASGPR engagement beyond multivalent glycan-based designs. Avilar Technologies has reported ASGPR-targeting chimeras (ATACs) that employ high-affinity small-molecule ASGPR ligands in a modular bispecific format to induce lysosomal degradation of extracellular targets, including TNFα, in HepG2 cells. In contrast to TGN-based approaches that rely on trivalent avidity, these molecules achieve efficient uptake using monovalent ASGPR-binding motifs with substantially improved intrinsic affinity, enabling target degradation at lower ligand stoichiometries and reduced molecular complexity. Together, these findings underscore that ASGPR engagement strength, achieved either through multivalency or enhanced ligand affinity, is a key determinant of degrader efficiency and represents an important design axis for next-generation extracellular degraders[75,76].

Third, a pronounced decrease in SPR sensorgram response with increasing TGN labeling suggests that steric hindrance may impair or even abolish IL-6 binding. This effect likely arises from modification of lysine residues in close proximity to the IL-6 interaction interface (e.g., K152 and K271), disrupting ligand engagement. Taken together, unfavorable kinetics, reduced binding affinity, and steric interference likely act in concert to limit target co-internalization by sIL-6R-based BioDegs.

Our overall observations support the notion that binding affinity alone is not the exclusive determinant of degrader efficacy. Instead, effective degradation depends on the formation of a favorable ternary complex, a principle that is well established for intracellular degraders such as PRO-TACs, where spatial configuration, cooperativity, and complex stability critically influence ubiquitination efficiency beyond binary binding

affinities[77]. Analogously, for TGN-based BioDegs, molecular size, scaffold geometry, and the stability of the target–degrader–ASGPR assembly appear to play decisive roles in enabling efficient internalization and lysosomal trafficking. Thus, similar to PROTAC systems, optimal degrader performance emerges from a balance of affinity, spatial organization, and complex dynamics rather than from maximal target binding alone.

Antibody modification represents a rapidly expanding area within biomedical research, offering opportunities to optimize the pharmacological properties of therapeutic antibodies and to confer novel biological functions. A clinically validated example of such engineering is exemplified by antibody-drug conjugates (ADCs), in which monoclonal antibodies are covalently linked to cytotoxic payloads to achieve selective eradication of antigen-expressing cells. Examples include trastuzumab-emtansine and trastuzumab-deruxtecan, both targeting HER2 but differing in linker design and payload, reflecting the tunable architecture of ADCs[78]. Beyond payload conjugation, various engineering strategies, such as site-specific modification, Fc glycoengineering, and the design of bispecific or multispecific antibodies, have become clinically established[79,80]. These modifications allow for enhanced selectivity, recycling, and functional diversity. The conceptual and technological framework established by ADC development provides a strong precedent for next-generation antibody-based modalities such as BioDegs, which similarly rely on modular antibody scaffolds for the selective engagement of both disease targets and lysosomal-targeting receptors. However, for BioDegs, the chemical conjugation of TGN or other receptor targeting motifs presents similar challenges as seen with ADCs. Non-site-specific chemical attachment can risk interfering with the antibody's antigen-binding affinity or alter its structural integrity. Therefore, more controllable and site-specific conjugation methods are crucial to ensure functional preservation and consistent performance. Chemoenzymatic strategies, such as sortase-mediated ligation or transglutaminase-catalyzed modification, offer promising approaches by enabling precise, reproducible coupling of receptor ligands at defined antibody sites[81,82]. These methods maintain the modularity and specificity required for efficient lysosomal targeting. As such, further development and optimization of these site-specific conjugation techniques remain a key area for advancing BioDeg technologies.

For glycan-based ASGPR-targeting degraders, a central translational challenge is dosing, because these agents act in a largely sacrificial manner: each degrader:cargo complex is internalized and routed through the endolysosomal pathway, effectively consuming the degrader along with its payload. As a result, sustained target clearance depends on maintaining sufficient systemic exposure over time to match ongoing target production and replenishment from the circulation. For high-abundance or rapidly resynthesized target proteins, this can translate into higher or more frequent dosing, which in turn raises practical constraints. A promising future direction for BioDeg evolution is the development of catalytic BioDegs, which retain the ability to bind, internalize, release, and re-engage multiple target molecules, analogous in function to sweeping antibodies or recycling antibodies[83,84]. These constructs are designed not for single-use degradation events, but for multiple target turnover, thereby significantly enhancing their therapeutic efficacy at lower doses. Recent preclinical concepts such as catalytic LYTACs (CataLYTACs) have demonstrated the feasibility of recycling degraders that exploit endogenous intracellular trafficking pathways, particularly those involving pH changes across endosomal compartments[66]. A key design principle in these systems is the pH-sensitive modulation of receptor binding affinity. Here, degraders are engineered to exhibit high-affinity binding to their lysosomal sorting receptor (e.g., ASGPR) as well as target antigens at physiological pH conditions, facilitating uptake at the cell surface, while exhibiting reduced affinity in the acidic environment of the endosome (~pH 5.5)[85]. This pH-triggered dissociation allows the degrader to release its cargo intracellularly, recycle back to the cell surface, and participate in subsequent rounds of target clearance. Kougentakis et al. demonstrated that glycan-based BioDegs with pH-selective target binding properties and improved glycan linker stabilities with respect to enzymatic breakdown can exploit FcRn-mediated recycling pathways,

achieving catalytic behavior through repeated endocytosis and release[66]. Such an approach would not only reduce the required dose or dosing frequency, but also align with the natural ligand recycling behavior of ASGPR, which is known to rapidly return to the plasma membrane after internalization. Given the modularity of the platform, glycan-based BioDegs could also be adapted for multispecific targeting, where different cytokines or immune mediators are addressed simultaneously, an especially relevant strategy in complex autoimmune and oncological contexts.

In summary, our study provides a close-meshed in vitro evaluation framework for early selection of BioDeg design concepts for targeting the IL-6 immune axis. For the further development of our most promising IL-6 and IL-6R targeting BioDeg variants, pre-clinical in vivo validation will be essential to assess pharmacokinetics, biodistribution, and therapeutic efficacy within the whole context of organismal immune systems.

## Methods
### Production of sASGPR and sIL-6R
sASGPR (ASGPR H1, with N-terminal Twin-Strep-Tag, 3xGS-Linker and TEV cleavage site, residues 62-291) and sIL-6R (1. with C-terminal His9-Tag, 2. with C-terminal Twin-Strep-Tag, 3xGS-Linker and TEV cleavage site; residues 20-355) were produced in Expi293F™ cells (Thermo Fisher Scientific). Cells were transfected using the ExpiFectamine™293 transfection kit according to the manufacturer's protocol. Briefly, cells were transfected with 1 μg/mL plasmid DNA in a total culture volume of 30 mL. Following transfection, cultures were incubated at 37 °C with 8% $CO_2$ for seven days. Cells were harvested by transferring the culture suspension into 50-mL Falcon tubes and centrifuging at $500 \times g$ for 5 min. The clarified supernatant was collected and stored at −20 °C until purification.

### Production of VHH against IL-6
Competent SHuffle® T7 (Express) E. coli were thawed on ice for 10 min. For each construct, 1 μL of purified plasmid DNA (10 ng) was added, mixed gently, and incubated on ice for 30 min. Heat shock was performed at 42 °C for 30 s, followed by 5 min on ice. After adding 950 μL SOC medium, cells were incubated at 30 °C with shaking at 300 rpm for 60 min. Aliquots of 20 μL and 200 μL were plated on selective agar (containing 100 μg/mL Ampicillin) and incubated overnight at the respective temperatures. A single colony was used to inoculate 5 mL LB with antibiotics and grown overnight at 37 °C, 180 rpm. Main cultures (100 mL LB with antibiotics) were inoculated 1:100 and grown at 37 °C until OD600 reached 0.5. Protein expression was induced with 0.2 mM IPTG and carried out at 30 °C for 4 h. Cells were harvested by centrifugation at $5000 \times g$ for 5 min at 4 °C, pellets were frozen at −20 °C.

### Production of siltuximab and tocilizumab
HEK293T cells were seeded at $7.2 \times 10^6$ cells per dish and cultured in growth medium at 37 °C with 5% $CO_2$ for 48 ho. At >70% confluency, cells were washed with 1x DPBS (containing $Ca^{2+}$ and $Mg^{2+}$), and the medium was replaced with fresh growth medium. Transfection complexes were prepared by mixing Lipofectamine 3000 and P3000 reagents with plasmid DNA encoding tocilizumab or siltuximab in Opti-MEM, followed by a 10-min incubation at room temperature. The mixture was then added dropwise to the cells, which were incubated for 4 h before supplementing with growth medium containing 25 mM HEPES buffer. Cultures were maintained at 37 °C and 5% $CO_2$, with supernatants collected regularly and replenished with fresh medium. From day 6 to 10, medium supplemented with HEPES was used. Supernatants from each day were frozen at −20 °C.

### Strep-Tactin® XT affinity chromatography
Twin-Strep-tagged proteins (sASGPR and sIL-6R) were purified via Strep-Tactin® XT affinity chromatography. Frozen culture supernatants were thawed at 37 °C and gently mixed. 10% (v/v) of 10x Strep-Tactin® XT buffer (IBA Lifesciences), supplemented with 1.5 mM EDTA and 0.025% BioLock (IBA Lifesciences), was added and incubated for 20 min at room temperature. The solution was centrifuged at 6000 x $g$ for 5 min and filtered

through 0.45 µm SFCA membranes. Affinity purification was performed using a 1 mL StrepTrap™ XT column (Cytiva) connected to an ÄKTA Start system. The column was equilibrated with 5 column volumes (CV) of MilliQ water followed by 5 CV of 1x Strep-Tactin® XT buffer containing 1.5 mM EDTA. Filtered supernatants were loaded at a flow rate of 1 mL/min. Flowthrough, wash fractions (8 CV of 1x buffer), and elution fractions (6 CV of Strep-Tactin® XT elution buffer) were collected for SDS-PAGE analysis. Relevant elution fractions were pooled for further processing (SEC).

### Nickel immobilized metal affinity chromatography (Ni-IMAC)
His-tagged proteins (sIL-6R and VHH against IL-6) were purified using a pre-packed 1 mL HisTrap column on an ÄKTA™ Start chromatography system. Prior to loading, samples and all buffers, including the Expi293F™ cell culture supernatant, were filtered. The column was initially washed with 10 column volumes (CV) of Milli-Q water and equilibrated with 5 CV of binding buffer (1x TBS + 20 mM imidazole, pH 7.4). The sample was then applied using 60 CV of binding buffer at a flow rate of 1 mL/min. Following sample application, the column was washed with 10 CV of binding buffer before elution was performed with 10 CV of elution buffer (1x TBS + 500 mM imidazole, pH 7.4) at the same flow rate. Fractions of 0.1 mL were collected throughout the flow-through, wash, and elution steps for quality assessment by SDS-PAGE. Elution fractions corresponding to chromatogram peaks were pooled for subsequent size exclusion chromatography.

### Protein A affinity chromatography
Supernatants from siltuximab and tocilizumab production were thawed at 37 °C for 30 min, pooled, and diluted with 10x PBS as indicated (e.g., 100 mL sample with 11 mL 10x PBS). The pooled samples were filtered through a 0.45 µm SFCA filter prior to purification. Purification was conducted on an ÄKTA™ Start system using a 1 mL HiTrap™ Protein A column. Tubing was primed before each step, with flow rates set to 60 rpm for priming and approximately 1 mL/min during purification. The column was washed with 5 column volumes (CV) of Milli-Q water, equilibrated with 10 CV of equilibration buffer (1x PBS), and the sample was applied. Post-application, the column was washed with 8 CV equilibration buffer. Elution was performed using 10 CV of elution buffer (100 mM sodium citrate, pH 3.5), with 50 µL of 1 M Tris pH 9.0 pre-added to collection tubes to neutralize fractions. All fractions (supernatant, wash, flow-through, elution) were collected for SDS-PAGE analysis. Elution fractions corresponding to chromatogram peaks were pooled for subsequent size exclusion chromatography.

### Size exclusion chromatography (SEC)
To enhance purity and remove aggregates, size exclusion chromatography was performed using an ÄKTA™ pure system equipped with a Superdex™ 200 Increase 10/300 GL column. All buffers were filtered and degassed prior to use. The column was initially washed with 3 column volumes (CV) of Milli-Q water, followed by 3 CV of 1x PBS buffer for equilibration. The sample (in 1x PBS) was injected via the sample loop and eluted with 1.5 CV of 1x PBS at an appropriate flow rate. Fractions of 0.5 mL were collected throughout elution and pooled according to chromatogram peaks. Purity was assessed by SDS-PAGE. Fractions containing the target protein were concentrated using an Amicon® Ultra centrifugal filter unit with a 10 kDa MWCO (sIL-6R and VHH) or 30 kDa MWCO (antibodies). Protein concentration was measured by absorbance at 280 nm using a NanoDrop™ UV spectrophotometer. Aliquots of purified protein were frozen in liquid nitrogen and stored at −20 °C for short-term or −80 °C for long-term storage.

### TGN conjugation
Labeling was performed by reacting purified proteins at a concentration >1 mg/mL with Tris-GalNAc-PEG5-sulfo-NHS Ester (Sussex Research, MV1000054, 5 mg/mL in DMSO) in 1x PBS. Molar ratios were chosen (1:1, 1:5, 1:10 and 1:25) based on available lysine residues. After overnight incubation at room temperature under rotation, reactions were quenched by adding Tris/HCl (pH 9.0) to 10 mM final concentration. Excess reagent was removed via three rounds of centrifugal filtration using Amicon® Ultra centrifugal filter units (Merck Millipore) with molecular weight cut-offs (MWCO) of 30 kDa for antibody-based BioDegs and 10 kDa for sIL-6R- and VHH-based constructs. Samples were washed three times with 300 µL of 1x PBS to remove unbound components and exchange buffer. Protein concentrations were determined using a NanoDrop™ UV-Vis spectrophotometer (Thermo Fisher Scientific).

### SDS-PAGE
SDS-PAGE was performed to confirm successful protein purification or modification and to quantify IL-6(R) levels in in vitro degradation assays. Prior to electrophoresis, samples were mixed with Laemmli loading buffer containing SDS. When required, 100 mM dithiothreitol (DTT) was added to ensure reducing conditions. Samples were then incubated at 95 °C for five min in a thermomixer. Subsequently, they were loaded onto precast 4-15% or 4–20% Mini-PROTEAN® TGX Stain-Free™ Protein Gels (Bio-Rad). Electrophoresis was carried out at 200 V for 45 min using the Bio-Rad PowerPac™ HC. Stain-free protein gel images were acquired using the ChemiDoc™ MP Imaging System (Bio-Rad) with activation for 45 s.

### Electrospray ionization mass spectrometry (ESI-MS)
The intact masses of proteins were determined by electrospray ionization mass spectrometry (ESI-MS) using a quadrupole time-of-flight (QTOF) mass spectrometer (Maxis, Bruker Daltonics) following desalting via high-performance liquid chromatography (HPLC, Agilent Technologies). A total of 200 µL of protein sample at a concentration of 0.25 µM was loaded onto a reversed-phase trapping column (0.8 × 2 mm, Poros R1; Applied Biosystems). After washing for 3 min with 0.3% formic acid at a flow rate of 300 µL/min, proteins were eluted using a mobile phase consisting of 40% isopropanol, 5% acetonitrile, and 0.3% formic acid at a flow rate of 40 µL/min.

Mass spectra were acquired in positive ion mode over an m/z range of 500–4000 using full-scan acquisition. The spectra rate was set to 2 Hz, and rolling average smoothing was applied with a 2× setting. Mass spectra were deconvoluted using the MaxEnt algorithm implemented in Compass DataAnalysis 4.2 software (Bruker Daltonics).

DAR values were calculated based on the peak intensity ratios of conjugated species, using the formula:

$$DAR = \sum DAR \text{ of the nth species x Peak intensity ratio of the nth species}$$

$$Peak\ intensity\ ratio\ of\ the\ nth\ species = \frac{Peak\ intensity}{Sum\ of\ peak\ intensities}$$

$$DAR\ red = 2(DAR\ LC + DAR\ HC)$$

The deconvoluted mass spectra were plotted using R[86] and Rpackage ggplot2[87].

### Thermal shift assay
Protein stability was assessed using a Prometheus NT.48 nanoDSF instrument (NanoTemper). Samples (0.5 mg/mL in PBS) were heated from 20–90 °C at 1 °C/min while monitoring intrinsic tryptophan fluorescence at 330 nm and 350 nm. Melting temperatures ($T_m$) were calculated from the inflection point of the 350/330 nm ratio curve using PR.StabilityAnalysis software (v2.0).

### Enzyme-linked immunosorbent assay (ELISA)
Binding assays were performed using defined buffer conditions: IL-6 and IL-6R binding experiments were carried out in 1x PBS (phosphate-buffered saline), while ASGPR binding assays were conducted in HBS-N buffer (10 mM HEPES, 150 mM NaCl, pH 7.4) supplemented either with 10 mM $CaCl_2$ to support calcium-dependent binding or with 10 mM EDTA to

chelate calcium and assess calcium-independent interactions. High-binding 96-well plates were coated overnight at 4 °C with 100 ng/well of IL-6, IL-6R, or ASGPR in the respective coating buffer. To block non-specific binding, wells were incubated with the corresponding assay buffer containing 5% BSA. Serial dilutions of BioDeg constructs were then added and incubated to allow binding. Detection of bound constructs was performed using HRP-conjugated secondary antibodies (Goat anti-human IgG HRP, A18811, Invitrogen, 1:1000; Mouse anti-His Tag HRP, MA180218, Invitrogen, 1:5000), followed by colorimetric development with TMB substrate. Between each step, wells were washed with the respective buffer supplemented with 0.05% Tween-20. The reaction was stopped using sulfuric acid, and absorbance was measured at 450 nm with background correction at 620 nm.

### Surface plasmon resonance (SPR) binding studies
For analyzing protein-protein and protein-glycan binding kinetics a Biacore T200 (Cytiva) system was used with CM5 sensor chips (Cytiva, Cat# BR100530). Ligands (ASGPR, IL-6 or IL-6R) were amine-coupled using an Amine Coupling Kit (Cytiva, Cat# BR100050): chips were activated with 1:1 EDC/NHS (7 min, 10 µL/min), followed by ligand injection (15 µg/mL in 10 mM sodium acetate, pH 5.0). Remaining active groups were blocked with 1 M ethanolamine (7 min).

For kinetic analysis, analytes (0.1–100 nM in HBS-N $++$ 10 mM CaCl$_2$) were injected at 30 µL/min for 180 s during the association step and 1200 s during the dissociation step. Data were reference-subtracted and fitted to a 1:1 binding model using Biacore Evaluation Software (v3.1). For interactions with a dissociation rate ($k_{off}$) below the instrument's detection limit, the $K_D$ was estimated using the measured association rate ($k_{on}$) and the theoretical maximum dissociation rate based on the applied duration for measuring off-rate kinetics.

### Cell culture and maintenance
HepG2 cells (DSMZ, ACC180) were maintained in Dulbecco's Modified Eagle Medium (DMEM, Gibco, Cat# 31885023) containing low glucose and pyruvate supplemented with 10% heat-inactivated fetal calf serum (FCS, Thermo Fisher, Cat# 10270106) and 1% penicillin/streptomycin (Thermo Fisher, Cat# 15140122). Cells were passaged at 80% confluence using Accutase® (Sigma-Aldrich, Cat# A6964). Cell counts and viability were assessed using trypan blue exclusion on a DeNovix CellDrop FL system. HEK-Blue™ IL-6 cells were maintained as adherent cultures in DMEM supplemented with 4.5 g/l glucose, 2 mM L-glutamine, 10% heat-inactivated fetal calf serum, 1% Penicillin-Streptomycin, 100 µg/ml Normocin™, and 1x HEK-Blue™ Selection reagent. Cells were subcultured when they reached 70–80% confluency and were not allowed to grow beyond 100% confluency. For passaging, cells were detached with pre-warmed PBS (37 °C). Growth medium was renewed twice per week. Test medium used for stimulation experiments was identical to the growth medium but lacked Normocin™ and HEK-Blue™ Selection.

### siRNA knockdown
siRNA-mediated knockdown of ASGPR H1 (*ASGR1*) was conducted using Lipofectamine® RNAiMAX. $7 \times 10^4$ cells/ well were transfected with 8 pmol siRNA/well in a 96-well format and incubated for 48 h with medium exchange after 24 h. Knockdown efficiency was assessed by flow cytometry, immunofluorescence and western blot.

### Flow cytometry for surface marker staining
Cells were seeded as described in the siRNA Knockdown section and harvested using Accutase®. Following collection, cells were washed with 1x PBS and incubated with Fc receptor blocking reagent (1:50 dilution in MACS buffer) for 15 min at 4 °C. Staining was performed in MACS buffer (1x PBS supplemented with 2 mM EDTA and 0.5% BSA). For detection of surface ASGPR, cells were incubated with a polyclonal anti-ASGPR primary antibody (rabbit anti-ASGR1, PA580356, Invitrogen, 1:1000 dilution in MACS buffer) for 30 min at 4 °C. After one wash with 150 µL MACS buffer, an

Alexa Fluor 488-conjugated anti-rabbit secondary antibody (A11008, Invitrogen, 1:1000 dilution) was applied for 30 min at 4 °C in the dark. To assess cell viability, Near-IR viability dye (1:1000 in 1x PBS) was added for 5 min at room temperature in the dark, followed by centrifugation at 500 x $g$ for 5 min. Cells were then washed twice with 150 µL MACS buffer (centrifuged at 500 x $g$ for 5 min each) and resuspended in 100 µL MACS buffer for acquisition. Flow cytometric analysis was performed on a MACSQuant® Analyzer 10 (Miltenyi Biotec), and data were analyzed using FlowJo software (version 10.1). A consistent gating strategy was applied across samples, including exclusion of debris, doublets, and dead cells (Supplementary Fig. 10).

### Western Blot analysis of protein level in HepG2 cells
A total of $7 \times 10^5$ cells were seeded in a 6-well plate and treated with siRNA as described above (80pmol siRNA/well). Cells were harvested using Accutase®, washed with PBS, and lysed by sonication in PBS supplemented with DNase I and protease inhibitor cocktail. Lysates were centrifuged to remove debris, and the resulting pellets were resuspended in Laemmli sample buffer. Samples were denatured at 95 °C for 5 min. Proteins were separated by SDS-PAGE and subsequently transferred to nitrocellulose membranes using the Trans-Blot Turbo Transfer System (Bio-Rad; 7 min, <25 V, 1.3 A). Membranes were rinsed with Milli-Q water and blocked with 5 mL of EveryBlot™ blocking buffer (Bio-Rad) for 5 mins at room temperature. ASGPR was detected using a polyclonal anti-ASGPR primary antibody (PA580356, Invitrogen, 1:1000 dilution in blocking buffer), followed by a CF680-conjugated secondary antibody (20418, Biotium, 1:10,000 dilution in blocking buffer). Total protein imaging via stain-free technology was used as a loading control.

### Immunofluorescence microscopy for surface marker staining
HepG2 cells ($1 \times 10^4$/well) were cultured in ibidi 8-well chambers. siRNA Knockdown was performed as described above with scaling of reagents to cell number. Following serum starvation (2 h, Opti-MEM®), cells were incubated with polyclonal anti-ASGPR antibody (PA580356, Invitrogen, 1:1000) in serum-free medium for 2 h. After fixation (4% paraformaldehyde, 20 min) and permeabilization (0.1% Triton X-100, 10 min), cells were blocked with 1% BSA and stained with an AF488-labeled secondary antibody (A11008, Invitrogen, 1:1000), Hoechst, and Phalloidin (1 h, RT, dark). Slides were mounted in Mowiol® mounting medium containing DABCO. Images were acquired using a Leica Stellaris confocal microscope equipped with HyD detectors. A 40x oil immersion objective was used. Fluorophores were excited with the following laser lines: 405 nm (Hoechst), 488 nm (AF488), and 561 nm (Phalloidin). Emission was detected using sequential line scanning. Image acquisition was controlled via Leica LAS X software (version 4.8.2.29567). Image processing was carried out using Fiji (ImageJ 1.54 g). All images were processed identically with only linear contrast and brightness adjustments applied to ensure direct comparability between experimental conditions.

### BioDeg uptake assay (flow cytometry)
For in vitro uptake assays, fluorescent labeling of BioDeg-target complexes was required for detection. For IL-6(R)-targeting constructs, labeling was achieved by pre-complexing biotinylated IL-6 or IL-6R with NeutrAvidin-Oregon Green 488 (Thermo Fisher Scientific), enabling fluorescence detection via receptor-mediated uptake. For sIL-6R variants containing a Twin-Strep-tag®, direct labeling was performed using fluorescent Strep-Tactin® XT DY-488 (IBA Lifesciences), allowing detection based on specific interaction with the tag. All flow cytometry and degradation experiments were carried out using three biological replicates, defined as independently thawed and cultured cell populations.

HepG2 cells were seeded in a 96-well plate, with or without siRNA treatment. After 48 h of incubation, the culture medium was removed, and cells were washed once with DPBS (with calcium and magnesium). Subsequently, 100 µL of Opti-MEM® serum-free medium was added per well, and cells were incubated for 2 h at 37 °C in a 5% CO$_2$ atmosphere. For

antibody-based BioDeg constructs, Fc receptor blocking was performed by adding 2 μL of FcR Blocking Reagent to 40 μL of medium per well, followed by incubation for 15 min at 37 °C and 5% $CO_2$. BioDeg samples were added at a final concentration of 100 nM in a total volume of 50 μL per well and incubated for 2 h at 37 °C and 5% $CO_2$. Following incubation, the medium was removed, and cells were washed twice with DPBS (-/-) supplemented with 2 mM EDTA. Cells were detached by adding 100 μL/well of Accutase® and incubated for 10 min at 37 °C. After harvesting, cells were stained for viability using Near-IR live/dead dye (1:1000 dilution in DPBS (-/-), 100 μL/well) for 5 min in the dark at room temperature. Cells were then centrifuged and washed twice with 200 μL of MACS buffer (PBS containing 2 mM EDTA and 0.5% BSA). The final cell pellet was resuspended in 100 μL of MACS buffer for flow cytometric analysis. Samples were analyzed using the MACSQuant® Analyzer 10 (Miltenyi Biotec), and data were processed using FlowJo software (version 10.1). A standardized gating strategy was applied to exclude debris, doublets, and dead cells (Supplementary Fig. 10).

### Lysosomal colocalization (live cell imaging)

HepG2 cells were seeded at a density of $1 \times 10^5$ cells/mL in 100 μL per well into 18-well ibidi μ-slides and cultured overnight. Uptake experiments were performed in phenol red-free medium consisting of low-glucose DMEM supplemented with 1x L-glutamine, 1% penicillin-streptomycin, 10% fetal calf serum (FCS), and 20 mM HEPES. Prior to sample addition, cells were incubated with Fc receptor blocking reagent (1:50 dilution) for 15 min at 37 °C and 5% $CO_2$. Sample uptake was carried out under conditions comparable to those described for flow cytometry. Immediately before imaging, 10 μL of staining solution (prepared by diluting LysoTracker™ [1:1000] and NucBlue™ [2 drops/mL] in culture medium) was added per well, followed by incubation for at least 10 min at 37 °C. Live-cell imaging was performed using a Leica Stellaris confocal microscope equipped with a 40x oil immersion objective. Fluorophores were excited with laser lines at 405 nm (NucBlue), 488 nm (Alexa Fluor 488), and 561 nm (LysoTracker), and emission was recorded using HyD detectors and sequential line scanning. Image processing and analysis were conducted in Fiji (ImageJ). All images were subjected to identical linear contrast adjustments via batch processing to ensure comparability across experimental conditions. Quantitative colocalization analysis was performed in Fiji (ImageJ 1.54 g) using the JACoP plugin, with manual thresholding applied before determining Pearson's correlation coefficient and Manders' overlap coefficient[88]. Manders' coefficient M1 was defined as the fraction of lysosomal signal (red channel) overlapping with the green channel (BioDeg or IL-6(R)), whereas M2 represented the fraction of green signal overlapping with the red channel. Colocalization was considered significant when M2 exceeded 0.5. To ensure statistical robustness, only images containing more than 10 cells were included in the analysis.

### Protein degradation analysis via western blot

Cells were seeded at a density of $1 \times 10^5$ cells/mL in 100 μL per well in a 96-well plate and incubated for 24 h at 37 °C and 5% $CO_2$. Samples (BioDeg plus respective target protein at 100 nM) were then added, and cells were incubated for an additional 24 h under the same conditions. Supernatants were collected and frozen for later use. Cells were lysed for 10 min with 50 μL 1x RIPA buffer containing protease inhibitors and 0.1 mg/mL DNase I, harvested, and transferred to Eppendorf tubes. Laemmli buffer was added to a final concentration of 1x, and samples were heated to 65 °C followed by sonication at 80% power for 5 s over 3 cycles. In a separate setup, after 24 h, supernatants were removed, cells washed with DPBS (+/+), and fresh medium was added for another 24 h before harvesting and processing as described. Total protein concentration in the 24-hour supernatant was determined by BCA assay, normalized across samples, mixed with Laemmli buffer, and heated at 65 °C for 10 min. Degradation of IL-6 and sIL-6R was analyzed via western blotting. Briefly, samples were seperated using SDS PAGE with subsequent transfer to nitrocellulose membranes using the Trans-Blot® Turbo system (Bio-Rad) with mixed MW protocol (25 V, 1.3 A, 7 min). Membranes were blocked with EveryBlot Blocking Buffer

(Bio-Rad) supplemented with BioLock for 15 min, probed with Strep-Tactin® XT DY-649 (1:1000) and anti-GAPDH CL594 (60004, Proteintech, 1:10,000) in Blocking Buffer for 1 h at RT. After washing with PBS-T, fluorescence signal was detected using the ChemiDocTM MP Imaging System (Bio-Rad).

### IL-6 activity assay using HepG2 and HEK-blue reporter cells

HepG2 cells were seeded at a density of $1 \times 10^5$ cells per well in 96-well flat-bottom plates and cultured for 24 h to allow adherence. Prior to stimulation, human interleukin-6 (IL-6) and interleukin-13 (IL-13) were prediluted to 1 μg/mL in PBS. Test compounds, including siltuximab, VHH_IL-6, soluble IL-6 receptor (sIL-6R), and their respective TGN-conjugated variants, were prepared at final concentrations of 0.4 nM, 4 nM, or 40 nM, each in combination with 100 ng/mL IL-6. All mixtures were preincubated at room temperature for 1 h to allow for complex formation. Cells were washed once with pre-warmed PBS containing calcium and magnesium (DPBS +/+) and incubated in 100 μL of serum-free Opti-MEM (Thermo Fisher Scientific) for 2 h at 37 °C and 5% $CO_2$ to induce serum starvation. Subsequently, the starvation medium was removed and cells were treated with 50 μL of the preincubated sample mixtures. Following 2 h of incubation under the same conditions, supernatants were collected for downstream analysis. HEK-Blue™ IL-6 reporter cells (InvivoGen) were harvested and resuspended at a density of $2.8 \times 10^5$ cells/mL in antibiotic-free test medium (DMEM high glucose medium supplemented with GlutaMAX, 1% penicillin/streptomycin and 10% FCS). For the reporter assay, 20 μL of HepG2 cell supernatant were transferred to a new 96-well plate and mixed with 180 μL of the HEK-Blue cell suspension (resulting in $5 \times 10^4$ cells/well). Cells were incubated for 20 h at 37 °C and 5% $CO_2$. To quantify SEAP (secreted embryonic alkaline phosphatase) activity, 180 μL of QUANTI-Blue™ reagent (InvivoGen), prepared according to the manufacturer's instructions, was combined with 20 μL of cell supernatant and incubated at 37 °C for 40 min. Absorbance was measured at 620 nm using a microplate spectrophotometer.

### Statistics and reproducibility

No statistical methods were used to predetermine sample sizes. Statistical analyses were performed as described in the respective figure legends. Exact n values, definitions of replicates, and details of statistical tests are provided in the corresponding sections of the manuscript and figure legends.

Differential scanning fluorimetry measurements using nanoDSF were performed in technical triplicates within one measurement. Surface plasmon resonance (SPR) and ELISA experiments were each performed once. Flow cytometry experiments and protein degradation experiments assessed by Western blot were performed with three independent biological replicates, defined as independently cultured HepG2 cells. A representative blot is shown in the figures. Live-cell imaging experiments for immunofluorescence microscopy were performed once per sample. Colocalization parameters (Manders' and Pearson's coefficients) were determined from a single acquisition per condition, with each image including at least 10 cells.

Data analysis was performed using GraphPad Prism 9 software. Data are expressed as means ± SD, and P values were determined by One-Way ANOVA/ Two-Way ANOVA.

### Reporting summary

Further information on research design is available in the Nature Portfolio Reporting Summary linked to this article.

### Data availability

All data generated or analyzed during this study are included in this published article and its Supplementary Information. All numerical source data underlying the graphs and charts are provided in the Supplementary Data file. Uncropped images of all blots and gels are available in Supplementary Fig. 11 and Fig. 12. The mass spectrometry proteomics data have been deposited to the ProteomeXchange Consortium via the PRIDE[89] partner repository with the dataset identifier PXD075514.

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

## Acknowledgements
We thank Christin Radon and Uwe Bierfreund from Cytiva for their technical support and helpful discussions. We are also grateful to Christian Müller and Marco Sisignano for their assistance with microscopy measurements. We further thank Anna Wiegand for her valuable support with ELISA experiments. This work was supported by the Innovation Centre Innovative Therapeutics (TheraNova) funded by the Fraunhofer Society, the Fraunhofer ATTRACT programme, the Hessian Ministry of Science and Research, Arts and Culture (HMWK), and the Nobel Laureate Fellowship of the Max Planck Society (to D.W.).

## Author contributions
A.E and M.S. conducted plasmid design. T.K., A.E., S.K. and A.S. produced and purified proteins. S.R. designed IF and FC uptake experiments and established protein production protocols. T.K. and A.E. developed methods for molecular interaction studies. D.W. provided support for structural analysis. A.K. secured funding and co-drafted the manuscript. M.L. and N.L. performed ESI-MS measurements and data analysis. M.S. performed protein production and modification, in vitro assays, analyzed and interpreted data, drafted the manuscript and prepared the figures. S.S. designed research, evaluated the data, secured funding, drafted the manuscript and prepared the figures.

## Funding

## Competing interests
Authors M.S, T.K, A.E, S.R, A.K and S.S are inventors on a provisional patent application related to assays employed in this work.
