## [Transparent Peer Review file · Communications Biology]

Glycan-based Biological Degradable Targeting the Cytokine Immune Axis

Corresponding Author: Dr Schara Safarian

Version 0:

Reviewer comments:

Reviewer #1

(Remarks to the Author)

"Glycan-based Biological Degradable Targeting the Cytokine Immune Axis" by Schara Safarian, Michelle Seifert, Tim Kollenkirchen, Andreas Ernst, Samaneh Rasoulinejad, Sarah Koellner, Di Wu, Aimo Kannt, and Ann-Kathrin Schneider (Manuscript Number: COMMSBIO-25-7705)

This manuscript described a glycan-based and asialoglycoprotein receptor (ASGPR)-mediated biological degradable (BioDegs) platform that took advantage of random lysine conjugation with triantennary N-acetylgalactosamine (TGN). By utilizing interleukin-6 (IL-6) and its soluble receptor (sIL-6R) as the model antigen, the authors systematically evaluated and compared a range of BioDeg formats. i.e. full-length antibodies (IgGs), VHH, and decoy-receptor with regard to TGN conjugation, binding affinity, thermal stability, cellular uptake, lysosomal trafficking, and degradation efficiency. The study concluded that the VHH-based BioDeg format appeared to be the best performing scaffold architecture despite exhibiting lower affinity and fewer TGN motifs than the full IgGs. They also found that high ASGPR binding might promote preferential internalization of ligand-free decoy degradable.

The research topic of this manuscript is interesting and the experimental design is well executed. The findings are original and relevant to the journal readers. While conjugating GalNAc to antibodies for lysosomal degradation is not a new technology, the authors were able to carry out a systematic comparison and characterization among different formats, and uncover some interesting mechanistic details. However, the following comments need to be clarified prior to the acceptance for publication:

- 1) Different protein-to-TGN ratios were utilized for random lysine conjugation, but the average TGN-to-protein ratios (Drug-Antibody Ratios) were not estimated for each conjugation condition. These DAR estimations should be provided, which could help to better interpret the ASGPR-binding results.
- 2) In Figure 2f, there is no detectable biacore binding with siltuximab-TGN (1:1) up to 200nM whereas the ELISA-based interaction is clearly visible in Figure 2e at 100nM. Which data is more trustworthy and why?

Minor points:

- 1) Reference #5 is the same as reference #33. Therefore the entire reference list should be rechecked carefully.
- 2) Chemical drawing for lysine conjugation should be provided in Figure 1 along with TGN for better experimental explanation.

Reviewer #2

(Remarks to the Author)

The study by Seifert et al. explores the promising field of targeted protein degradation (TPD) within modern drug discovery. Specifically, the authors harness the endolysosomal pathway to degrade proteins of interest (POIs), in this case, interleukin-6 (IL-6) and its soluble receptor (sIL-6R). Similar to other lysosome-targeting chimera (LYTAC) strategies, their approach relies on asialoglycoprotein receptors (ASGPR). What distinguishes this work are two novel aspects: the focus on IL-6 and sIL-6R as targets, and the comparison of three different binding modalities, full-length antibodies, VHH fragments, and a decoy receptor design.

The manuscript begins with an overview of the TPD concept, including examples of intracellular degradation such as PROTACs, before transitioning to extracellular approaches like LYTACs and relevant therapeutic targets. The results are organised according to the binding modality used for IL-6 or sIL-6R, with each section employing consistent biophysical techniques (nanoDSF, SPR, and ELISA-based assays) to evaluate the synthesised degraders (BioDegs). During synthesis, the authors varied the ratio of protein binder to ASGPR binder, producing distinct reaction outcomes.

For in vitro studies, they used the ASGPR-expressing liver cell line HepG2 and generated a knockdown variant via siRNA. Uptake experiments were confirmed by flow cytometry and immunoblotting, while confocal fluorescence microscopy (CFM) combined with lysosomal markers was used to monitor internalisation and assess co-localisation with lysosomes – a key requirement for degradation.

The best-performing BioDegs from each binder class were then compared. The authors found that the choice of protein-binding modality significantly influenced degradation performance and, importantly, that binding affinity did not directly correlate with degradation efficiency.

Major concerns:

What remains insufficiently addressed is the conceptual novelty of this work in relation to existing ASGPR-targeting LYTACs. The manuscript does not clearly explain the rationale for selecting IL-6 and sIL-6R as degradation targets in liver cells, nor does it justify the choice of full-length antibodies, VHHs, and decoy receptors – particularly in terms of their respective advantages and why these approaches merit direct comparison.

The comparison of the best-performing BioDegs is presented in a way that does not optimally support the stated goal of identifying the most suitable IL-6/sIL-6R-binding modality.

Moreover, the discussion section requires substantial revision. At present, it reads more like a general literature review than a critical analysis of the study's findings in relation to existing knowledge. Greater integration of the experimental results with the broader protein degradation landscape is needed to strengthen the discussion.

Overall, another major concern is that the authors claim to have established a pipeline for early evaluation of LYTAC development. However, as it is well known in the field of ASGPR targeting and drug delivery in general, the main obstacles occur in vivo and without such comparison the results presented here can only serve as a preliminary and early prioritization tool. If in vivo work would confirm the predictability of the results, this would be a different case. This should also be included in the discussion.

Finally, the degree of functionalization should be quantified using MALDI.

Section- and figure-specific comments:

Abstract/Scheme 1:

- • What are Biological Degradator (BioDegs) unclear where the name comes from?
- • Triantennary N-acetylgalactosamine (TGN) is the same ligand for ASGPR as before: by Ahn et al., Nat. Chem. Biol. 2021, 17, 937–946 (DOI: 10.1038/s41589-021-00770-1) antibody-tri-GalNAc conjugates (GalNAc-LYTACs)
- • Novelty are the targets: IL-6, sIL-6R, and the antigen-binding units full-length antibodies, VHH, and binding decoy-receptor

Introduction:

- • p.1, line 39: “A fundamental distinction between molecular degraders and conventional inhibitors or neutralizing proteins is that MDs do not require to interact with functionally important sites to disrupt biochemical processes. Instead, degraders operate by recognizing accessible surfaces of pathogenic proteins, regardless of their biochemical activity.”
- o please provide references, this information is available in the common literature about PROTACs, but most of the successful warheads are actually high-affinity inhibitors.
- o Also, what is the author's take on this information for your LYTAC design?
- • P.2, line 50: “antibodies. This concept was pioneered by Igawa et al. in 2010, who demonstrated how pH-dependent binding of an engineered anti-IL-6R IgG1 can facilitate IL-6R degradation”.
- o It is good to mention this literature, but the authors do not do pH-dependent binding engineering and their BioDegs do not rely on Fcn-receptor for recycling; please formulate in a way to better connect it to your research.
- • The relevance of degrading IL-6 and sIL-6R is not highlighted; is it of use to degrade the cytokine and its soluble receptor in hepatocytes?
- • The rationale of using the three different POI binding moieties is unclear, what are the expectation when using them?
- • The scope of the study sounds vague, please refine.
- • Fig. 1: the chemical structure contains many structural errors.
- o Please double check! (You show galactosamine not GalNAc, also the stereo centers of the carbohydrate do not correspond to IUPAC drawing of a carbohydrate and should be revised etc.)

Results:

Siltuximab-TGN: Degradation of IL-6 via a Monoclonal Antibody:

- • Suppl. Fig. 1:
- o Why is there IL-2 sig. in the plasmid map?
- • Line 109 ff.: what is the conclusion from surface plasmon resonance (SPR) measurements? Are the results indicative for the performances
- • Line 118: Does “sASGPR” here refer to the literature-known hetero-oligomeric complex sASGPR, the secreted form of H1 and H2 of ASGPR? – please clarify to avoid confusion.
- • It is likely that the sASGPR is expressed as a monomer. On cells the ASGPR can exist as a heterotrimer, this should be mentioned to allow the reader to interpret the results.
- • Line 126 ff.: this is already known “These findings confirm the specific and calcium-dependent interaction between TGN moieties and the galactose-binding site of ASGPR” – what you mean is that upon attachment to siltuximab they still bind to

sASGPR, please doublecheck

- Figure 2: comparing e and f: for 1:10 vs. 1:25 there is same behaviour in 100 nM, while for f 1:25 looks distinctly different. Do you have any explanation?
- Figure 2: it is unclear how dissociation constants are derived if koff cannot be determined and slow kon does not lead to reaching equilibrium. If these data are not reliable, then they should not be reported. However, overall it is apparent that TGN modification results in reduced on-rates.
- Along the same lines: How was the SPR analysis done if the MW of the e.g. antibody was not defined?
- Figure 3:
 - o Why WB (a) and CFM (c) have not been quantified to compare it with FC (b)?
 - o Apparently all the techniques to verify silencing efficacy have been applied. In (c) neg. ctrl. siRNA would be missing then.
 - o In (e) x-axis "488 nM"?
 - o Why mean fluorescence intensity is used for flow cytometry as some distributions look like skewed distributions? – It is better to use the median for skewed distributions.
- The siRNA targeting the ASGPR, was it against H1 or H2? Please indicate.

Anti-IL-6 VHH-TGN: A Camelid Immunoglobulin-Based BioDeg targeting IL-6:

- Motivation for the use of VHH is unclear, especially when the number of reactive lysine groups for coupling is 4 and for full-length IgG it has been shown that high coupling ratio performs better.
- Fig. 6 (f): What can be seen at VHH_IL-6? – please comment
- Line 242: "while unmodified VHH and 242 TGN ratios 1:1 and 1:5 showed weak uptake signals presumably attributable to ASGPR-independent 243 mechanisms (Fig. 7c)."
 - o Or due to insufficient ASGPR silencing?
- Figure 8 shows loss of binding of sIL-6R-TGN upon high labeling. Then Figure 9 still shows internalization. This should be explained.

sIL-6R-TGN: Decoy Receptor-Based Targeting of IL-6:

- Fig. 10: caption, (f) before (e).
- The authors claim to assess the importance of modulating target binding as well as receptor engagement: in the presented way the target binding can only insufficiently decoupled from the influence of the TGN since its coupling (valency, presentation and chemical modification of the target scaffold) is different for each. Still, the absolutely best performing BioDeg can be identified from these data and it is just strongly recommend to adjust the scope of the comparison.

Discussion:

- The discussion section has been used for introduction to the topic which is not the actual use of the discussion section. Please focus on the discussion of your Results and comparison to the relevant literature.
- Relevance of the POIs?
- Line 391: "TGN modification of the antibody- and VHH-based constructs did not interfere with their neutralizing properties in an IL-6 reporter assay, confirming preserved biological activity upon conjugation"
 - o acc. to the common sense about concept of degraders, neutralising activity is not necessarily required from them to be active. Please indicate why you highlight this as an important property?
- Line 411 ff.: paragraph sounds rather speculative, please revise.
- Line 428ff.: paragraph, this is known for degraders which rely on ternary complex formation, similar like for PROTACs, see: [10.1016/j.chembiol.2017.09.010](https://doi.org/10.1016/j.chembiol.2017.09.010), please revise.
- Overall, the discussion should be significantly shortened. The first two paragraphs of the discussion read like the introduction and either these additional information are directly put into context with the results of the study or they should be moved into the introduction. The discussion should be shortened significantly as a rational design cannot be inferred from this study without in vivo data. The paragraph about CatalYTACs is not necessary.

Reviewer #3

(Remarks to the Author)

Summary

The manuscript "Glycan-based Biological Degradation Targeting the Cytokine immune Axis" by Michelle Seifert and others, describes the development of a biologic modality for the targeted degradation of proteins via lysosomal function. Here, the authors modified anti-IL6 antibodies and a soluble IL6 Receptor (sIL6R) with triantennary N-acetylglucosamine (TGN) to elicit interactions with the asialoglycoprotein receptor (ASGPR) and drive cellular uptake and degradation. The authors show a thorough analytical and functional analyses that TGN modified targeting agents can bind and target IL6 for degradation in vitro. Further, by assessing the clearance rate and level of IL6 using different constructs (IgGs, VHH and sIL6R) the authors show that the activity is not dependent solely on affinity and TGN labeling, but also the size and architecture of the targeting agent. Overall, the manuscript is well written, the analysis is thorough and appropriate and provides interesting insights into the mechanisms behind ASGPR dependent clearance.

General comments

The manuscript provides a very well written and presented analysis of TGN modified antibodies as clearance vehicles and use the cytokine IL6 as their test bed. IL6 is an important proinflammatory cytokine with clear therapeutic potential in autoimmunity and cancer. The manuscript is well presented, well written and the data supports the conclusions drawn by the authors. The data itself provides interesting mechanistic information regarding the use of ASGPR as a means of targeted

degradation. Indeed, the authors demonstrate that ASPGR targeting with TGN can elicit efficient degradation and cytokine clearance in vitro. The authors present the right controls to demonstrate the specificity of the mechanism and provide multiple lines of evidence to support their model. By comparing different modalities of the targeting agents (IgG, VHH and sIL6R) the authors demonstrate that while affinity and the extent of TGN labeling are important, the size and architecture of the vehicle plays a big role in driving efficacy.

Major points:

As written the manuscript is complete and accurate. It is of interest to the emerging use of targeted degraders for a number of medical needs. While this work does report interesting results in a very well controlled system, it is restricted to an in vitro setting. Greater impact could be gained by assessing these constructs in an in vivo setting and observing their potential to control disease. The manuscript could greatly improve if the authors investigated the effect of biodistribution, PK and on target efficacy. While this work is important to understanding the mechanisms behind ASPGR dependent clearance, its true value lies in the application to medicine. This work is a great first step in establishing TGN labeled molecules for protein clearance, but in practice the extended pharmacokinetics of IgGs (for example) may offer significant benefits to performance to outweigh increase clearance rates seen in vitro. The testing of these molecules in vivo would provide a great step up in the importance of this work.

Minor points:

As mentioned above the manuscript is extremely well written and discussed. The reviewer would suggest the following minor edits:

1. Introduction: Discuss the importance of IL6 clearance to human health and disease to put this work into context. While this point is covered in the discussion it would be beneficial to the readers to understand why IL6 was selected and why it is important to medicine.
2. Line 159: Correct the misspelling of (Fig. 3e)
3. Line 219: The ratios TGN:Ab (10:1) are consistently reported as Ab:TGN throughout the text and figures. While not incorrect I would suggest switching it for consistency.
4. Paragraph starting on line 289: This section is absolutely correct, but could benefit from the explicit clarification that figure 9 demonstrates the internalization of sIL6R. Upon first read the reviewer was confused for why Fig9 and Fig10 show different results. Both figures use the sIL6R-IL6 complex and it required closer examination to reveal that Fig9 follows the sIL6R internalization.

Reviewer #4

(Remarks to the Author)

The manuscript presents a systematic comparison of antibody-, VHH-, and receptor-based glycan conjugates for targeted degradation of IL-6 and sIL-6R. The study is timely, well executed, and highly relevant to the emerging field of extracellular protein degraders.

1. While lysosomal trafficking is inferred, the use of lysosomal inhibitors (e.g., chloroquine, bafilomycin A1) could confirm degradation pathway dependence. Have the authors tested whether blocking lysosomal acidification alters degrader efficacy?
2. The discussion mentions catalytic BioDegs and pH-sensitive modifications as future directions. Could the authors elaborate on the key challenges that need to be addressed for clinical translation, such as stability in circulation, immunogenicity of glycan modifications, or dosing strategies?
3. The manuscript shows lysosomal co-localization of targets, but it is unclear whether the BioDeg scaffolds themselves are degraded or recycled. Have the authors attempted to label the degrader and track its nature?
4. A consolidated table summarizing KD values, uptake efficiencies, and degradation percentages for each of the developed degraders would help readers directly compare performance.
5. The manuscript shows lysosomal co-localization, but it remains unclear whether endosomal sorting efficiency differs among the degraders. Have the authors examined early vs. late endosome markers to track trafficking kinetics?
6. The authors noted difficulty in fitting multivalent TGN-ASGPR interactions with a 1:1 binding model. Have the alternative kinetic models (heterogeneous ligand, avidity-corrected, or bivalent analyte models) been tested to capture the complexity of these interactions better?
7. The discussion references sweeping antibodies and therapeutic precedents (e.g., omalizumab, LCA-0061) but does not consistently cite primary clinical trial or preclinical data sources. Adding references to original reports strengthens the statement.
8. The manuscript mentions that the results of the chimeric anti-IL-6 siltuximab antibody were recombinantly produced in HEK293T cells and purified using Protein A affinity chromatography followed by size exclusion chromatography. However, the Experimental section does not provide sufficient detail on this procedure (e.g., plasmid constructs, transfection parameters, culture conditions, Protein A elution buffers, and SEC column specifications). I recommend that the authors include a dedicated subsection in the Methods section that describes the antibody expression and purification workflow in detail.

Reviewer #5

(Remarks to the Author)

I co-reviewed this manuscript with one of the reviewers who provided the listed reports. This is part of the Communications Biology initiative to facilitate training in peer review and to provide appropriate recognition for Early Career Researchers who co-review manuscripts.

Version 1:

Reviewer comments:

Reviewer #1

(Remarks to the Author)

The manuscript is much improved and acceptable for publication.

Reviewer #2

(Remarks to the Author)

All comments have been addressed.

Reviewer #3

(Remarks to the Author)

The changes and additions made by the authors satisfy my concerns. This is an impactful study.

Reviewer #4

(Remarks to the Author)

Upon reviewing the authors' responses to the previous comments and the corresponding revisions made to the manuscript. Overall, the authors have addressed the scientific and methodological concerns satisfactorily, and the revisions have improved the clarity and completeness of the work. The added discussion points, methodological clarifications, and summary figure strengthen the manuscript and align well with the scope of the study. I therefore recommend acceptance of the manuscript for publication in its current form.

Reviewer #5

(Remarks to the Author)

I co-reviewed this manuscript with one of the reviewers who provided the listed reports. This is part of the Communications Biology initiative to facilitate training in peer review and to provide appropriate recognition for Early Career Researchers who co-review manuscripts.

Reviewer comments:

Reviewer #1 (Remarks to the Author):

“Glycan-based Biological Degradable Targeting the Cytokine Immune Axis” by Schara Safarian, Michelle Seifert, Tim Kollenkirchen, Andreas Ernst, Samaneh Rasoulinejad, Sarah Koellner, Di Wu, Aimo Kannt, and Ann-Kathrin Schneider (Manuscript Number: COMMSBIO-25-7705)

This manuscript described a glycan-based and asialoglycoprotein receptor (ASGPR)-mediated biological degraders (BioDegs) platform that took advantage of random lysine conjugation with triantennary N-acetylgalactosamine (TGN). By utilizing interleukin-6 (IL-6) and its soluble receptor (sIL-6R) as the model antigen, the authors systematically evaluated and compared a range of BioDeg formats. i.e. full-length antibodies (IgGs), VHH, and decoy-receptor with regard to TGN conjugation, binding affinity, thermal stability, cellular uptake, lysosomal trafficking, and degradation efficiency. The study concluded that the VHH-based BioDeg format appeared to be the best performing scaffold architecture despite exhibiting lower affinity and fewer TGN motifs than the full IgGs. They also found that high ASGPR binding might promote preferential internalization of ligand-free decoy degraders.

The research topic of this manuscript is interesting and the experimental design is well executed. The findings are original and relevant to the journal readers. While conjugating GalNAc to antibodies for lysosomal degradation is not a new technology, the authors were able to carry out a systematic comparison and characterization among different formats, and uncover some interesting mechanistic details. However, the following comments need to be clarified prior to the acceptance for publication:

1) Different protein-to-TGN ratios were utilized for random lysine conjugation, but the average TGN-to-protein ratios (Drug-Antibody Ratios) were not estimated for each conjugation condition. These DAR estimations should be provided, which could help to better interpret the ASGPR-binding results.

We thank the reviewer for bringing up this very important aspect regarding the DAR determination. We have now performed additional intact mass spectrometry studies to estimate the average glycan-to-protein ratio (GPR) for each conjugation condition. While we have been successful in determining the GPR for our IgG and VHH variants, the sIL-6R variants did not allow for DAR estimation due to highly heterogeneous glycosylation of this protein. We have specified this limitation in the manuscript text. All novel GPR data have been added to the revised manuscript and discussed within the context of our data. (see Supplementary Figure 5).

2) In Figure 2f, there is no detectable biacore binding with siltuximab-TGN (1:1) up to 200nM whereas the ELISA-based interaction is clearly visible in Figure 2e at 100nM. Which data is more trustworthy and why?

We thank the reviewer for highlighting this important point. The apparent discrepancy between the Biacore and ELISA data arises from fundamental differences in the respective assay formats. Biacore (SPR) provides a real-time, label-free measurement of binding kinetics under continuous flow, whereas ELISA reports end-point binding after prolonged incubation under near-equilibrium conditions.

Several factors can therefore lead to differing apparent sensitivities between ELISA and SPR for the same interaction pair. Most notably, ELISA benefits from enzymatic signal amplification and typically involves high surface densities of immobilized ligand, both of which can enhance the detectability of weak or transient interactions. In the specific case of the data shown in Fig. 2e, an additional effect is likely contributing to the observed difference in detection threshold. The high surface enrichment of sASGPR in the ELISA format, combined with the multivalent nature of the TriGalNAc moiety, is expected to promote avidity effects and rebinding, thereby increasing apparent sensitivity. These effects are substantially reduced in the flow-based SPR experiment, where analyte:ligand contact times are shorter, and binding is assessed under more stringent kinetic conditions. Given these fundamental differences between the two methodologies a direct correlation of results is not possible.

With respect to the question of confidence in one method over the other, it is important to recognize that both approaches offer complementary strengths while also having inherent limitations. SPR enables resolution of binding kinetics and allows for quantitative determination of dissociation constants, whereas ELISA provides high analytical sensitivity and facilitates detection of target interactions even at very low protein amounts.

In the present study, ELISA clearly demonstrates that binding of TGN-modified antibodies to sASGPR is calcium-dependent, thereby substantiating that the interaction is mediated by the specific lectin activity of sASGPR. In addition, we employ both ELISA and SPR to provide orthogonal evidence that binding of TGN-modified Siltuximab variants depends on both labeling stoichiometry and antibody concentration. Together, these data support the specificity of the interaction and demonstrate consistent trends across two conceptually distinct experimental methodologies.

Minor points:

1) Reference #5 is the same as reference #33. Therefore the entire reference list should be rechecked carefully.

We thank the reviewer for pointing this out. The reference list has been carefully rechecked, and the duplication has been removed. All references have now been renumbered and verified for accuracy.

2) Chemical drawing for lysine conjugation should be provided in Figure 1 along with TGN for better experimental explanation.

We thank the reviewer for this valuable suggestion. A schematic representation of the lysine conjugation reaction, including the TGN structure, has now been added to Figure 1 in the revised manuscript to improve the clarity of the experimental workflow.

Reviewer #2 (Remarks to the Author):

The study by Seifert et al. explores the promising field of targeted protein degradation (TPD) within modern drug discovery. Specifically, the authors harness the endolysosomal pathway to degrade proteins of interest (POIs), in this case, interleukin-6 (IL-6) and its soluble receptor (sIL-6R). Similar to other lysosome-targeting chimera (LYTAC) strategies, their approach relies on asialoglycoprotein receptors (ASGPR). What distinguishes this work are two novel aspects: the focus on IL-6 and sIL-6R as targets, and the comparison of three different binding modalities, full-length antibodies, VHH fragments, and a decoy receptor design.

The manuscript begins with an overview of the TPD concept, including examples of intracellular degradation such as PROTACs, before transitioning to extracellular approaches like LYTACs and relevant therapeutic targets. The results are organised according to the binding modality used for IL-6 or sIL-6R, with each section employing consistent biophysical techniques (nanoDSF, SPR, and ELISA-based assays) to evaluate the synthesised degraders (BioDegs). During synthesis, the authors varied the ratio of protein binder to ASGPR binder, producing distinct reaction outcomes.

For in vitro studies, they used the ASGPR-expressing liver cell line HepG2 and generated a knockdown variant via siRNA. Uptake experiments were confirmed by flow cytometry and immunoblotting, while confocal fluorescence microscopy (CFM) combined with lysosomal markers was used to monitor internalisation and assess co-localisation with lysosomes – a key requirement for degradation.

The best-performing BioDegs from each binder class were then compared. The authors found that the choice of protein-binding modality significantly influenced degradation performance and, importantly, that binding affinity did not directly correlate with degradation efficiency.

Major concerns:

What remains insufficiently addressed is the conceptual novelty of this work in relation to existing ASGPR-targeting LYTACs. The manuscript does not clearly explain the rationale for selecting IL-6 and sIL-6R as degradation targets in liver cells, nor does it justify the choice of full-length antibodies, VHHs, and decoy receptors – particularly in terms of their respective advantages and why these approaches merit direct comparison. The comparison of the best-performing BioDegs is presented in a way that does not optimally support the stated goal of identifying the most suitable IL-6/sIL-6R-binding modality.

We thank the reviewer for highlighting this point and helping us in improving the quality of our manuscript. We have clarified motivation and the rationale behind this study by modifying the introduction section and providing a clearer rationale behind the choice of the degradation strategy and the molecules tested in head-to-head manner.

Moreover, the discussion section requires substantial revision. At present, it reads more like a general literature review than a critical analysis of the study's findings in relation to existing knowledge. Greater integration of the experimental results with the broader protein degradation landscape is needed to strengthen the discussion.

We thank the reviewer for this suggestion. While we prefer to maintain the references to the broad literature, we acknowledge that there is lack of integration of our findings in the broader context of immune biology and TPD. Hence, we have revised the discussion paragraph to focus on critical interpretation of our experimental results in the context of extracellular protein

degraders and LYTAC strategies. The revised discussion now emphasizes how our findings on BioDeg scaffold architecture, receptor engagement, and degradation efficiency contribute to the broader understanding of extracellular protein degradation and highlight potential directions for future development.

Overall, another major concern is that the authors claim to have established a pipeline for early evaluation of LYTAC development. However, as it is well known in the field of ASGPR targeting and drug delivery in general, the main obstacles occur *in vivo* and without such comparison the results presented here can only serve as a preliminary and early prioritization tool. If *in vivo* work would confirm the predictability of the results, this would be a different case. This should also be included in the discussion.

We fully agree that *in vivo* studies are essential to evaluate pharmacokinetics, biodistribution, and therapeutic efficacy. The current study was designed to provide a controlled and systematic *in vitro* approach for evaluation and early prioritization of BioDeg design variants. We have clarified in the revised discussion that our study reflects a closed-meshed evaluation procedure on the biochemical, biophysical and *in vitro* levels, and that *in vivo* validation for gaining in-depth pre-clinical insights into the potency of design variants in the context of inflammatory cytokine targeting is an essential downstream step marking a significant milestone for drug candidate identification (line 560 ff.).

Finally, the degree of functionalization should be quantified using MALDI.

We agree that determination of DAR ratios is an important metric for assessing and comparing the biochemical and *in vitro* outcomes of our study. We have performed additional intact mass spectrometry studies to estimate the average DAR for each conjugation condition. While we have been successful in determining the glycan-to-protein ratios (GPR) for our IgG and VHH variants, the sIL6R variant did not allow for DAR estimation due to highly heterogeneous glycosylation of this protein. We have mentioned this limitation in the manuscript text. All novel GPR data have been added to the revised manuscript and discussed within the context of our biochemical and *in vitro* data. Please note that we report GPR instead of DAR describing glycan-to-protein ratio (line 137 ff., 236 ff., 282 ff., 323 ff.).

Section- and figure-specific comments:

Abstract/Scheme 1:

- What are Biological Degradator (BioDegs) unclear where the name comes from?

We introduce the term 'Biological Degradators (BioDegs)' to coin a broader term for molecules that induce target degradation which are based on biological frameworks.

- Triantennary N-acetylgalactosamine (TGN) is the same ligand for ASGPR as before: by Ahn et al., Nat. Chem. Biol. 2021, 17, 937–946 (DOI: 10.1038/s41589-021-00770-1) antibody-tri-GalNAc conjugates (GalNAc-LYTACs). Novelty are the targets: IL-6, sIL-6R, and the antigen-binding units full-length antibodies, VHH, and binding decoy-receptor

Our abstract now explicitly states IL-6, sIL-6R, and comparison of binder modalities as novel aspects.

Introduction:

- p.1, line 39: “A fundamental distinction between molecular degraders and conventional inhibitors or neutralizing proteins is that MDs do not require to interact with functionally important sites to disrupt biochemical processes. Instead, degraders operate by recognizing accessible surfaces of pathogenic proteins, regardless of their biochemical activity.”
 - please provide references, this information is available in the common literature about PROTACs, but most of the successful warheads are actually high-affinity inhibitors.
 - Also, what is the author’s take on this information for your LYTAC design?

We have added the respective reference for this claim (Madan et al., 2022).

While functional modulation of a target protein is technically not required for degrader design, provided that accessible and sufficiently high-affinity binding sites can be engaged, functional sites may in practice offer chemically and structurally favorable interaction surfaces. Accordingly, the identification of suitable binding sites for mechanistically effective degraders is likely to depend on multiple factors, including local protein dynamics, charge distribution, and spatial configuration compatible with either ubiquitination (for PROTACs) or receptor engagement and endocytic trafficking (for BioDegs).

- P.2, line 50: “antibodies. This concept was pioneered by Igawa et al. in 2010, who demonstrated how pH-dependent binding of an engineered anti-IL-6R IgG1 can facilitate IL-6R degradation”.
 - It is good to mention this literature, but the authors do not do pH-dependent binding engineering and their BioDegs do not rely on Fcn-receptor for recycling; please formulate in a way to better connect it to your research.

We thank the reviewer for recognizing our careful citation of highly relevant work from other laboratories. We would like to clarify that the purpose of this section of the Introduction is to provide a concise, chronological overview of key breakthrough technologies in the area of extracellular biological degraders. While not all examples discussed are directly related to the present study, we consider this broader perspective valuable, particularly for non-expert readers, as it helps contextualize our work within the wider field of targeted protein degradation and biologically based modalities.

- The relevance of degrading IL-6 and sIL-6R is not highlighted; is it of use to degrade the cytokine and its soluble receptor in hepatocytes?

We thank the reviewer for highlighting the lack of clarity in this matter. We have clarified the rationale behind hepatocyte-specific clearance of IL6 and sIL6R in our revised introduction paragraph.

In brief: IL-6 is a key inflammatory cytokine implicated in chronic inflammatory diseases, autoimmune disorders, and cancer. Its soluble receptor (sIL-6R) extends the cytokine’s signaling range via trans-signaling. Hepatocytes are highly relevant target cells because the liver expresses ASGPR abundantly, allowing efficient lysosomal uptake. IL-6 is in numerous diseases systemically upregulated, making targeted degradation in the liver particularly rational.

Targeted degradation of IL-6 and sIL-6R in hepatocytes thus represents a proof-of-concept for extracellular degrader efficacy in physiologically relevant cells.

- The rationale of using the three different POI binding moieties is unclear, what are the expectations when using them?

We have clarified that the three binder modalities (full-length antibodies, VHH fragments, and decoy receptor) were chosen to systematically explore how binder size, valency, and epitope recognition influence ASGPR-mediated uptake and degradation. Full-length antibodies offer high valency and well-characterized pharmacokinetics, VHH fragments provide small size with high binding affinity, and decoy receptors allow mimicking natural protein-protein interactions without the need for drug discovery efforts. The revised Introduction now explicitly outlines the properties and the rationale for testing these three modalities (line 119 ff.).

- The scope of the study sounds vague, please refine.

We thank the reviewer for the comment. We have revised the Introduction to clarify the scope of our study (line 114 ff.).

- Fig. 1: the chemical structure contains many structural errors.
 - Please double check! (You show galactosamine not GalNAc, also the stereo centers of the carbohydrate do not correspond to IUPAC drawing of a carbohydrate and should be revised etc.)

We have carefully revised Figure 1 to correct all chemical representation issues. The structure now depicts triantennary N-acetylgalactosamine (GalNAc) correctly, with accurate stereochemistry according to IUPAC conventions. All previous errors, including the misrepresentation of galactosamine and incorrect stereocenters, have been corrected.

Results:

Siltuximab-TGN: Degradation of IL-6 via a Monoclonal Antibody:

- Suppl. Fig. 1:
 - Why is there IL-2 sig. in the plasmid map?

The IL-2 signal represents a secretion signal peptide used for secretory production of siltuximab in HEK293 expression cell lines.

- Line 109 ff.: what is the conclusion from surface plasmon resonance (SPR) measurements? Are the results indicative for the performances

In conclusion, SPR measurements indicate that TGN modification preserves IL-6 binding, with only slightly slower association rates at higher modification levels, suggesting that target engagement and overall high-affinity are largely maintained.

- Line 118: Does “sASGPR” here refer to the literature-known hetero-oligomeric complex sASGPR, the secreted form of H1 and H2 of ASGPR? – please clarify to avoid confusion.

- It is likely that the sASGPR is expressed as a monomer. On cells the ASGPR can exist as a heterotrimer, this should be mentioned to allow the reader to interpret the results.

In our study, sASGPR refers to the monomeric extracellular domain H1 used in SPR. We now clarify in the text that cellular ASGPR exists as a heterotrimer, and that monomeric sASGPR is used as a surrogate to assess ligand binding *in vitro* (line 165 ff.).

- Line 126 ff.: this is already known “These findings confirm the specific and calcium-dependent interaction between TGN moieties and the galactose-binding site of ASGPR” – what you mean is that upon attachment to siltuximab they still bind to sASGPR, please doublecheck

We revised the text to clarify the conclusion derived from these sets of experiments (line 173 ff.).

- Figure 2: comparing e and f: for 1:10 vs. 1:25 there is same behaviour in 100 nM, while for f 1:25 looks distinctly different. Do you have any explanation?

We thank the reviewer for highlighting this. The apparent discrepancy between the Biacore and ELISA data most likely arises from fundamental differences in the respective assay formats. Biacore (SPR) provides a real-time, label-free measurement of binding kinetics under continuous flow, whereas ELISA reports end-point binding after prolonged incubation under near-equilibrium conditions. Several factors can therefore lead to differing apparent sensitivities between ELISA and SPR for the same interaction pair. Most notably, ELISA benefits from enzymatic signal amplification and typically involves high surface densities of immobilized ligand, both of which can enhance the detectability of weak or transient interactions. In the specific case of the data shown in Fig. 2e, an additional effect is likely contributing to the observed difference in detection threshold. The high surface enrichment of sASGPR in the ELISA format, combined with the multivalent nature of the TriGalNAc moiety, is expected to promote avidity effects and rebinding, thereby increasing apparent sensitivity. These effects are substantially reduced in the flow-based SPR experiment, where analyte:ligand contact times are shorter, and binding is assessed under more stringent kinetic conditions.

In Fig. 2e, signals at 100 nM approach saturation, which may reflect a combined effect of high TriGalNAc avidity at both labeling conditions and signal amplification associated with HRP-based detection. This observation illustrates the importance of assessing binding across a range of protein concentrations, as measurements performed closer to the linear response regime are more informative for resolving relative differences in binding behavior.

- Figure 2: it is unclear how dissociation constants are derived if k_{off} cannot be determined and slow k_{on} does not lead to reaching equilibrium. If these data are not reliable, then they should not be reported. However, overall it is apparent that TGN modification results in reduced on-rates.

We thank the reviewer for this comment. The dissociation rate constant (k_{off}) was too slow to be accurately measured, yet we can still derive that k_{off} is not faster than $1 \times 10^{-5} \text{ s}^{-1}$.

The association rate constant (k_{on}) on the other hand could be reliably determined. Using these values in a 1:1 binding model, we calculated an apparent K_D as reported. This approach provides an upper bound for K_D while confirming that TGN modification reduces the on-rate. The Methods sections have been updated to clarify this interpretation (Line 1141 ff.).

- Along the same lines: How was the SPR analysis done if the MW of the e.g. antibody was not defined?

The SPR experiments were performed using molar analyte concentrations determined by UV absorbance at 280 nm applying the Beer–Lambert law. Extinction coefficients were calculated based on the primary amino-acid sequence of IL-6(R)-binding proteins, which is identical for all conjugates. Glycan moieties do not contribute to absorbance at 280 nm, as they lack aromatic chromophores. Consequently, the A_{280} -based quantification reflects the concentration of protein molecules rather than total mass. While glycosylation increases molecular weight, this does not affect the determination of molar concentration used for SPR binding analyses, which depend on the number of binding-competent protein molecules rather than their mass.

- Figure 3:
 - Why WB (a) and CFM (c) have not been quantified to compare it with FC (b)?
 - Apparently all the techniques to verify silencing efficacy have been applied. In (c) neg. ctrl. siRNA would be missing then.
 - In (e) x-axis “488 nM”?
 - Why mean fluorescence intensity is used for flow cytometry as some distributions look like skewed distributions? – It is better to use the median for skewed distributions.
 - The siRNA targeting the ASGPR, was it against H1 or H2? Please indicate.

In this instance we chose WB and CFM data as qualitative readouts. We chose to employ flow cytometry as the quantitative method to assess statistical significance of the knockdown. In our opinion, the presented data clearly demonstrates successful silencing of *ASGPR H1* expression.

The negative control siRNA was not included in the immunofluorescence experiments, as its lack of effect on *ASGPR1* expression was already validated by two independent methods beforehand (WB and flow cytometry).

The x-axis labeling error “488 nM” has been corrected.

For consistency with the related literature, mean fluorescence intensity was used to report flow cytometry data, rather than median values.

The siRNA used in our experiments specifically targets ASGPR H1. We have updated this information in the methods section (line 1161 ff.).

Anti-IL-6 VHH-TGN: A Camelid Immunoglobulin-Based BioDeg targeting IL-6:

- Motivation for the use of VHH is unclear, especially when the number of reactive lysine groups for coupling is 4 and for full-length IgG it has been shown that high coupling ratio performs better.

We clarified that VHH fragments were chosen for their small size, ease of production, and potential for rapid tissue penetration, despite fewer lysines (line 271 ff.).

The general motivation for including a VHH-based binder was obviously not to maximize TGN valency, but to systematically probe how protein framework architecture influences BioDeg performance independent of coupling capacity. VHHs represent a compact and rigid binding modality with favorable tissue penetration and reduced conformational flexibility compared to full-length IgGs. Despite offering fewer reactive lysine residues for TGN conjugation, the VHH-based BioDeg achieved efficient ASGPR-mediated uptake and IL-6 degradation, in some cases outperforming IgG-based constructs. We regard these novel data as highly important and useful, as these insights uncover that high TGN stoichiometry is not the sole determinant of BioDeg efficacy.

- Fig. 6 (f): What can be seen at VHH_IL-6? – please comment

The SPR sensorgram of VHH_IL-6 shows only minor transient responses upon injection. Between 0–500 s, small positive and negative deviations from baseline are observed, likely caused by bulk refractive index changes rather than specific binding. After 500 s, the signal returns to approximately 0 RU, indicating a non-stable interaction with the surface. These initial fluctuations can result from non-specific adsorption or desorption, small differences in buffer composition or temperature, and changes in flow rate. Overall, the sensorgram reflects the typical behavior of a non-binding analyte, where only brief injection artifacts are visible, followed by a baseline return.

- Line 242: “while unmodified VHH and 242 TGN ratios 1:1 and 1:5 showed weak uptake signals presumably attributable to ASGPR-independent 243 mechanisms (Fig. 7c).”
 - Or due to insufficient ASPGPR silencing?

We thank the reviewer for this comment. No ASGPR silencing was performed in the live cell imaging experiments but not in this set of experiments. This line references data derived from wildtype HepG2 cells. The weak signal observed for unmodified VHH and VHH-TGN ratios (1:1 and 1:5) was similar to the control (as seen in our Fc data) and is therefore likely due to ASGPR-independent uptake.

- Figure 8 shows loss of binding of sIL-6R-TGN upon high labeling. Then Figure 9 still shows internalization. This should be explained.

Figure 9 shows the uptake of sIL-6R alone. Therefore, the reduced binding to IL-6 observed in Figure 8 at high labeling levels is not relevant in this context, as internalization occurs independently of IL-6 binding. The section has been reformulated to clarify this point (line 347 ff.).

sIL-6R-TGN: Decoy Receptor-Based Targeting of IL-6:

- Fig. 10: caption, (f) before (e).

The figure caption has been corrected.

- The authors claim to assess the importance of modulating target binding as well as receptor engagement: in the presented way the target binding can only insufficiently decoupled from the influence of the TGN since its coupling (valency, presentation and chemical modification of the target scaffold) is different for each. Still, the absolutely best performing BioDeg can be identified from these data and it is just strongly recommend to adjust the scope of the comparison.

We thank the reviewer for this critical and helpful comment. Within the rationale section in the last paragraph of the introduction section we have adjusted the scope and goal of this study to highlight that our overall aim is to systematically evaluate how combined differences in target-binding modality and ASGPR engagement influence BioDeg performance in the context of IL-6 immune axis targets (line 114 ff.).

Discussion:

- The discussion section has been used for introduction to the topic which is not the actual use of the discussion section. Please focus on the discussion of your Results and comparison to the relevant literature.

We thank the reviewer for this comment. We have ensured to substantially shorten the first paragraphs of the discussion section to shift focus on the interpretation of our results and their implications in the context of existing studies and biology.

- Relevance of the POIs?

We clarified the clinical and biological relevance of the protein targets IL-6 and sIL-6R. IL-6 is a key inflammatory cytokine, and sIL-6R extends its signaling range via trans-signaling. Hepatocytes are relevant target cells due to high ASGPR expression. Since IL-6 is often elevated systemically in various diseases, targeting it in the liver is particularly meaningful. These points demonstrate why degradation of these targets is biologically relevant in the context of ASGPR-mediated protein clearance. This information has been added to the Introduction to better contextualize the choice of POIs (line 83 ff.).

- Line 391: “TGN modification of the antibody- and VHH-based constructs did not interfere with their neutralizing properties in an IL-6 reporter assay, confirming preserved biological activity upon conjugation”
 - acc. to the common sense about concept of degraders, neutralising activity is not necessarily required from them to be active. Please indicate why you highlight this as an important property?

We thank the reviewer for this comment. From a methodological point of view the use of the signaling reporter assay serves as an orthogonal approach to validate whether conjugation of siltuximab or VHH have affected target-binding. Thus, in the broader context of our study, we use this technique primarily as a means for deeper characterization of the tested modalities.

However, the general question on whether neutralization is a benefit (disruption of biological activity) or disadvantage (more available epitope regions) when it comes to biological degrader design, has motivated us to add a more in-depth and multilayered discussion paragraph regarding this point (line 426 ff.).

- Line 411ff.: paragraph sounds rather speculative, please revise.

Speculative statements were removed or rewritten to provide a more evidence-based interpretation of our results.

- Line 428ff.: paragraph, this is known for degraders which rely on ternary complex formation, similar like for PROTACs, see: [10.1016/j.chembiol.2017.09.010](https://doi.org/10.1016/j.chembiol.2017.09.010), please revise.

We thank the reviewer for helping us improve the overall quality of our study. We have revised this paragraph and made a reference to similarities in the PROTAC field citing the suggested literature (line 498 ff.).

- Overall, the discussion should be significantly shortened. The first two paragraphs of the discussion read like the introduction and either these additional information are directly put into context with the results of the study or they should be moved into the introduction. The discussion should be shortened significantly as a rational design cannot be inferred from this study without in vivo data. The paragraph about CatalYTACs is not necessary.

The Discussion has been shortened substantially, with introductory material moved to the Introduction where appropriate.

Reviewer #3 (Remarks to the Author):

Summary

The manuscript “Glycan-based Biological Degradable Targeting the Cytokine immune Axis” by Michelle Seifert and others, describes the development of a biologic modality for the targeted degradation of proteins via lysosomal function. Here, the authors modified anti-IL6 antibodies and a soluble IL6 Receptor (sIL6R) with triantennary N-acetylglucosamine (TGN) to elicit interactions with the asialoglycoprotein receptor (ASGPR) and drive cellular uptake and degradation. The authors show a thorough analytical and functional analyses that TGN modified targeting agents can bind and target IL6 for degradation in vitro. Further, by assessing the clearance rate and level of IL6 using different constructs (IgGs, VHH and sIL6R) the authors show that the activity is not dependent solely on affinity and TGN labeling, but also the size and architecture of the targeting agent. Overall, the manuscript is well written, the analysis is thorough and appropriate and provides interesting insights into the mechanisms behind ASGPR dependent clearance.

General comments

The manuscript provides a very well written and presented analysis of TGN modified antibodies as clearance vehicles and use the cytokine IL6 as their test bed. IL6 is an important proinflammatory cytokine with clear therapeutic potential in autoimmunity and cancer. The manuscript is well presented, well written and the data supports the conclusions drawn by the authors. The data itself provides interesting mechanistic information regarding the use of ASGPR as a means of targeted degradation. Indeed, the authors demonstrate that ASPGR targeting with TGN can elicit efficient degradation and cytokine clearance in vitro. The authors present the right controls to demonstrate the specificity of the mechanism and provide multiple lines of evidence to support their model. By comparing different modalities of the targeting agents (IgG, VHH and sIL6R) the authors demonstrate that while affinity and the extent of TGN labeling are important, the size and architecture of the vehicle plays a big role in driving efficacy.

Major points:

As written the manuscript is complete and accurate. It of interest to the emerging use of targeted degraders for a number of medical needs. While this work does report interesting results in a very well controlled system, it is restricted to an in vitro setting. Greater impact could be gained by assessing these constructs in an in vivo setting and observing their potential to control disease. The manuscript could greatly improve if the authors investigated the effect of biodistribution, PK and on target efficacy. While this work is important to understanding the mechanisms behind ASPGR dependent clearance, its true value lies in the application to medicine. This work is a great first step in establishing TGN labeled molecules for protein clearance, but in practice the extended pharmacokinetics of IgGs (for example) may offer significant benefits to performance to outweigh increase clearance rates seen in vitro. The testing of these molecules in vivo would provide a great step up in the importance of this work.

We thank the Reviewer for her/his encouraging and constructive comments. We fully agree that *in vivo* evaluation, including pharmacokinetics and biodistribution for this novel modality class, as well as on-target efficacy, is crucial towards clinical translation. The current study was

designed to establish a fundamental framework for evaluating design principles and the systematic testing of glycan-based degraders covering sample analytics (nanoDSF/LC-MS), profiling of target/receptor interactions (ELISA/SPR), as well as uptake and degradation analyses (FC/IF/WB) in a controlled *in vitro* system.

We have clarified this scope and rationale in the revised Discussion section, emphasizing that future studies will aim to extend these findings to *in vivo* models to evaluate pharmacokinetic behavior and therapeutic potential. (line 560 ff.)

Minor points:

As mentioned above the manuscript is extremely well written and discussed. The reviewer would suggest the following minor edits:

1. Introduction: Discuss the importance of IL6 clearance to human health and disease to put this work into context. While this point is covered in the discussion it would be beneficial to the readers to understand why IL6 was selected and why it is important to medicine.

We thank the reviewer for this valuable suggestion. We have revised the Introduction to include a paragraph discussing the medical relevance of IL6 and its pathological role in inflammatory and autoimmune diseases. This addition clarifies the rationale behind selecting IL6 as the model system for this study (line 83 ff.).

2. Line 159: Correct the misspelling of (Fig. 3e)

We thank the reviewer for pointing out the typographical error. The misspelling of "(Fig. 3e)" has been corrected.

3. Line 219: The ratios TGN:Ab (10:1) are consistently reported as Ab:TGN throughout the text and figures. While not incorrect I would suggest switching it for consistency.

We thank the reviewer for the suggestion regarding ratio notation. The ratios in line 256 have been adjusted and are now consistently described as Ab:TGN throughout the manuscript and figures for clarity and consistency.

4. Paragraph starting on line 289: This section is absolutely correct, but could benefit from the explicit clarification that figure 9 demonstrates the internalization of sIL6R. Upon first read the reviewer was confused for why Fig9 and Fig10 show different results. Both figures use the sIL6R-IL6 complex and it required closer examination to reveal that Fig9 follows the sIL6R internalization.

We thank the reviewer for this insightful comment. We have revised the corresponding paragraph to explicitly clarify that Figure 9 demonstrates the internalization of sIL6R, whereas Figure 10 shows the uptake of the sIL6R-IL6 complex. This clarification has been added to prevent potential confusion (line 347 ff.).

Reviewer #4 (Remarks to the Author):

The manuscript presents a systematic comparison of antibody-, VHH-, and receptor-based glycan conjugates for targeted degradation of IL-6 and sIL-6R. The study is timely, well executed, and highly relevant to the emerging field of extracellular protein degraders.

1. While lysosomal trafficking is inferred, the use of lysosomal inhibitors (e.g., chloroquine, bafilomycin A1) could confirm degradation pathway dependence. Have the authors tested whether blocking lysosomal acidification alters degrader efficacy?

We appreciate this suggestion and agree that lysosomal inhibition experiments represent a valuable approach to mechanistically validate endo-lysosomal degradation pathways. In the present study, however, our primary objective was to establish a procedural and comparative framework for evaluating different IL-6- and sIL-6R-binding protein scaffolds functionalized with TriGalNAc moieties, focusing on their biochemical and biophysical properties, and *in vitro* cellular performances.

TriGalNAc-mediated ASGPR engagement is a well-characterized and extensively validated targeting strategy, widely used for hepatocyte-specific delivery of siRNA and has been shown to induce receptor-mediated endocytosis and degradation. Prior studies employing TriGalNAc-based or GalNAc-derived ASGPR ligands have convincingly demonstrated lysosome-dependent degradation using pharmacological inhibitors such as bafilomycin A1 or chloroquine (doi: <https://doi.org/10.1016/j.chembiol.2022.12.003>; [10.1021/acscentsci.1c00146](https://doi.org/10.1021/acscentsci.1c00146); <https://doi.org/10.1021/jacs.5c13607>, etc.). In light of consistent body of literature, we consider lysosomal routing to be an established property of TriGalNAc-functionalized constructs and therefore did not prioritize repeating such validation experiments within the scope of the current work.

2. The discussion mentions catalytic BioDegs and pH-sensitive modifications as future directions. Could the authors elaborate on the key challenges that need to be addressed for clinical translation, such as stability in circulation, immunogenicity of glycan modifications, or dosing strategies?

We thank the reviewer for this comment. We have added an additional paragraph which references challenges in dosing and consideration for clinical translation. We kept this paragraph brief, as we were advised to shorten the discussion paragraph substantially (line 531 ff.).

3. The manuscript shows lysosomal co-localization of targets, but it is unclear whether the BioDeg scaffolds themselves are degraded or recycled. Have the authors attempted to label the degrader and track its nature?

We thank the reviewer for this insightful question. The interaction of ASGPR with GalNAc-containing ligands is well established to be pH-dependent, reflecting the receptor's physiological role in scavenging desialylated glycoproteins (DOI: [10.1074/jbc.274.50.35400](https://doi.org/10.1074/jbc.274.50.35400)). Upon endocytosis, ligand dissociation is promoted by endosomal acidification, enabling ASGPR recycling to the cell surface, while internalized cargo is routed toward lysosomal compartments. This pH-dependent lectin behavior has been demonstrated not only for endogenous ASGPR ligands but also for engineered GalNAc- and TriGalNAc-based targeting

moieties, as the underlying carbohydrate–lectin interaction chemistry is conserved (DOI: 10.1038/s41589-021-00770-1). While not of special interest for our current study, our results regarding the internalization and lysosomal trafficking of sIL-6R corroborate the pH-dependent lectin property of TriGalNAc. (Fig. 9).

4. A consolidated table summarizing KD values, uptake efficiencies, and degradation percentages for each of the developed degraders would help readers directly compare performance.

We fully agree with this helpful suggestion. A summary figure has been added to the discussion section, providing an overview of the key quantitative parameters (KD, uptake efficiency, and degradation percentage) for each degrader construct. This addition facilitates direct comparison and enhances the clarity of the manuscript (Figure 11).

5. The manuscript shows lysosomal co-localization, but it remains unclear whether endosomal sorting efficiency differs among the degraders. Have the authors examined early vs. late endosome markers to track trafficking kinetics?

We thank the reviewer for this insightful comment. In the current study, we did not examine early versus late endosomal trafficking and therefore cannot comment on potential differences in endosomal sorting efficiency among the degrader types. We agree that such analyses would provide valuable mechanistic insight and have highlighted this as an important direction for future studies in the revised Discussion. (line 457 ff.)

6. The authors noted difficulty in fitting multivalent TGN–ASGPR interactions with a 1:1 binding model. Have the alternative kinetic models (heterogeneous ligand, avidity-corrected, or bivalent analyte models) been tested to capture the complexity of these interactions better?

We thank the reviewer for this valuable comment. The multivalent TGN–ASGPR conjugates are inherently heterogeneous, consisting of a mixture of molecules with different valencies. Due to these effects of valency and intrinsic heterogeneity, the Biacore sensorgrams cannot be accurately described using available kinetic models, including heterogeneous ligand or bivalent analyte models. As a result, fitting the data to extract kinetic rate constants would not represent good practice. Hence, we chose to only report the sensorgram information as evidence for conjugation-stoichiometry dependent binding, but not to include kinetic evaluation to avoid setting a bad precedent.

7. The discussion references sweeping antibodies and therapeutic precedents (e.g., omalizumab, LCA-0061) but does not consistently cite primary clinical trial or preclinical data sources. Adding references to original reports strengthens the statement.

Relevant pre-clinical studies and clinical trials have been cited for sweeping and recycling antibodies (Tulika et al., 2024; Yamamura et al., 2019). For LCA-0061, only a preprint is available and has been cited accordingly. Additional clinical studies for Omalizumab have been included (Busse et al., 2001; Maurer et al., 2013), which specifically address the changes in IgE levels following treatment.

8. The manuscript mentions that the results of the chimeric anti-IL-6 siltuximab antibody were recombinantly produced in HEK293T cells and purified using Protein A affinity chromatography

followed by size exclusion chromatography. However, the Experimental section does not provide sufficient detail on this procedure (e.g., plasmid constructs, transfection parameters, culture conditions, Protein A elution buffers, and SEC column specifications). I recommend that the authors include a dedicated subsection in the Methods section that describes the antibody expression and purification workflow in detail.

We thank the reviewer for this suggestion. The recombinant expression and purification of siltuximab, including plasmid constructs, transfection parameters, culture conditions, Protein A affinity chromatography, and SEC specifications, are described in detail in the Methods section (Line 1011 ff, 1046 ff., 1059 ff) and shown in supplementary figures 1-4.